# How Much is Unseen Depends Chiefly on Information About the Seen

**Seongmin Lee and Marcel Böhme**
MPI for Security and Privacy, Germany
{seongmin.lee,marcel.boehme}@mpi-sp.org

## Abstract

The *missing mass* refers to the proportion of data points in an *unknown* population of classifier inputs that belong to classes *not* present in the classifier's training data, which is assumed to be a random sample from that unknown population. We find that *in expectation* the missing mass is entirely determined by the number $f_k$ of classes that *do* appear in the training data the same number of times *and an exponentially decaying error*. While this is the first precise characterization of the expected missing mass in terms of the sample, the induced estimator suffers from an impractically high variance. However, our theory suggests a large search space of nearly unbiased estimators that can be searched effectively and efficiently. Hence, we cast distribution-free estimation as an optimization problem to find a distribution-specific estimator with a minimized mean-squared error (MSE), given only the sample. In our experiments, our search algorithm discovers estimators that have a substantially smaller MSE than the state-of-the-art Good-Turing estimator. This holds for over 93% of runs when there are at least as many samples as classes. Our estimators' MSE is roughly 80% of the Good-Turing estimator's.

## 1 Introduction

How can we extrapolate from properties of the training data to properties of the unseen, underlying distribution of the data? This is a fundamental question in machine learning (Orlitsky et al., 2003; Orlitsky & Suresh, 2015; Painsky, 2022; Acharya et al., 2013; Hao & Li, 2020). The probability that a data point belongs to a class that does *not* exist in the training data is also known as the *missing probability mass* since *empirically* the entire probability mass is distributed over classes that *do* exist in the training data. For instance, the missing mass measures how *representative* the training data is of the unknown distribution. If the missing mass is high, the training is not very representative, and a trained classifier is unlikely to predict the correct class. If we manually label training data, the missing mass also measures *saturation*. We may decide that the labeling effort has been sufficient and saturation has been reached when the missing mass is below a certain threshold.

### 1.1 Background

Consider a *multinomial distribution* $p = \langle p_1, \cdots p_S \rangle$ over a support set $\mathcal{X}$ where support size $S = |\mathcal{X}|$ and probability values are *unknown*. Let $X^n = \langle X_1, \cdots X_n \rangle$ be a set of independent and identically distributed random variables representing the sequence of elements observed in $n$ samples from $p$. Let $N_x$ be the number of times element $x \in \mathcal{X}$ is observed in the sample $X^n$. For $k : 0 \leq k \leq n$, let $\Phi_k$ be the number of elements appearing exactly $k$ times in $X^n$, i.e., $N_x = \sum_{i=1}^n \mathbf{1}(X_i = x)$ and $\Phi_k = \sum_{x \in \mathcal{X}} \mathbf{1}(N_x = k)$. Let $f_k(n)$ be the expected value of $\Phi_k$ (Good, 1953), i.e.,

$$f_k(n) = \binom{n}{k} \sum_{x \in \mathcal{X}} p_x^k (1 - p_x)^{n-k} = \mathbb{E}[\Phi_k] \tag{1}$$

**Estimating rare/unobserved** $p_x$. We cannot expect all elements to exist in $X^n$. While the empirical estimator $\hat{p}_x^{\text{Emp}} = N_x/n$ is generally unbiased, $\hat{p}_x^{\text{Emp}}$ distributes the entire probability mass only over

the observed elements. This leaves a "missing probability mass" over the unobserved elements. In particular, $\hat{p}_x^{\text{Emp}}$ *given that* $N_x > 0$ overestimates $p_x$, i.e., for observed elements

$$\mathbb{E}\left[\frac{N_x}{n} \;\middle|\; N_x > 0\right] = \frac{p_x}{1 - (1 - p_x)^n}. \tag{2}$$

We notice that the bias increases as $p_x$ decreases. Bias is maximized for the rarest observed element.

**Missing mass, realizability, and natural estimation**. Good and Turing (GT) (Good, 1953) discovered that the expected value of the probability $M_k = \sum_{x \in \mathcal{X}} p_x \mathbf{1}(N_x = k)$ that the $(n + 1)$-th observation $X_{n+1}$ is an element that has been observed exactly $k$ times in $X^n$ (incl. $k = 0$) is a function of the expected number of colors $f_{k+1}(n + 1)$ that will be observed $k + 1$ times in an enlarged sample $X^n \cup X_{n+1}$, i.e, $\mathbb{E}[M_k] = \frac{k+1}{n+1} f_{k+1}(n + 1)$. We also call $M_k$ as *total probability mass* over the elements that have been observed exactly $k$ times. Since our sample $X^n$ is only of size $n$, GT suggested to estimate $M_k$ using $\Phi_{k+1}$. Concretely, $\hat{M}_k^G = \frac{k+1}{n} \Phi_{k+1}$.

For $k = 0$, $M_0$ gives the *"missing" (probability) mass* over the elements not in the sample. In genetics and biostatistics, the complement $1 - M_0$ measures *sample coverage*, i.e., the proportion of individuals in the population belonging to a species *not* observed in the sample (Chao & Jost, 2012). In the context of supervised machine learning, assuming the training data is a random sample, the sample coverage of the training data gives the proportion of all data (seen or unseen) with labels not observed in the training data.

A *natural estimator* of $p_x$ assigns the same probability to all elements $x$ appearing the same number of times in the sample $X^n$ (Orlitsky & Suresh, 2015). For $k > 0$, $\hat{p}_x = M_k/\Phi_k$ gives the hypothetical *best natural estimator* of $p_x$ for every element $x$ that has been observed $k$ times.

**Bias of GT**. In terms of bias, Juang & Lo (1994) observe that the GT estimator $\hat{M}_k^G = \frac{k+1}{n} \Phi_{k+1}$ is an unbiased estimate of $M_k(X^{n-1})$, i.e., where the $n$-th sample was *deleted* from $X^n$ and find:

$$\left|\mathbb{E}\left[\hat{M}_k^G - M_k\right]\right| = \left|\mathbb{E}\left[M_k(X^{n-1}) - M_k(X^n)\right]\right| \leq \frac{k+2}{n+1} = \mathcal{O}\left(\frac{1}{n}\right). \tag{3}$$

**Convergence/competitiveness of GT**. McAllester & Schapire (2000) analyzed the *convergence*, which is then improved by Drukh & Mansour (2004) and more recently by Painsky (2022). They showed that, with high probability, $\hat{M}_k^G$ converges at a rate of $\mathcal{O}(1/\sqrt{n})$ for all $k$ based on worst-case mean squared error analysis. Using the Poisson approximation, Orlitsky & Suresh (2015) showed that natural estimators from GT, i.e., $\hat{p}_x^G = \hat{M}_{N_x}^G/\Phi_{N_x}$, performs close to the best natural estimator. Regret, the metric of the competitiveness of an estimator against the best natural estimator, is measured as KL divergence between the estimate $\hat{p}$ and the actual distribution $p$, $D_{KL}(\hat{p}||p)$. Their study also showed that finding the best natural estimator for $p$ is same as finding the best estimator for $M = \{M_k\}_{k=0}^n$.

**Poisson approximation**. The Poisson approximation with parameter $\lambda_x = p_x n$ has often been used to tackle a major challenge in the formal analysis of the missing mass and natural estimators (Orlitsky & Suresh, 2015; Orlitsky et al., 2016; Acharya et al., 2013; Efron & Thisted, 1976; Valiant & Valiant, 2016; Good, 1953; Good & Toulmin, 1956; Hao & Li, 2020). The challenge is the *dependencies between frequencies* $N_x$ for different elements $x \in \mathcal{X}$. In this Poisson Product model, a continuous-time sampling scheme with $S = |\mathcal{X}|$ independent Poisson distributions is considered where the frequency $N_x$ of an element $x$ is represented as a Poisson random variable with mean $p_x n$. In other words, the frequencies $N_x$ are modelled as independent random variables. Consequently, the GT estimator is unbiased in the Poisson Product model (Orlitsky et al., 2016); yet, it is biased in the multinomial distribution (Juang & Lo, 1994). Hence, we tackle the dependencies between frequencies analytically, without approximation via the Poisson Product model.

## 1.2 Contribution of the Paper

In this paper, we reinforce the foundations of multinomial distribution estimation with a precise characterization of the *dependencies* between $N_x = \sum_{i=1}^n \mathbf{1}(X_i = x)$ across different $x \in \mathcal{X}$ (rather than assuming independence using the Poisson approximation). The theoretical analysis is based on the *expected value* of the frequency of frequencies $\mathbb{E}[\Phi_k] = f_k(n)$ between different $k$ and $n$, which is

$$\frac{f_k(n)}{\binom{n}{k}} = \frac{f_k(n+1)}{\binom{n+1}{k}} - \frac{f_{k+1}(n+1)}{\binom{n+1}{k+1}}. \tag{4}$$

Exploring this new theoretical tool, we bring two contributions to the estimation of the total probability mass $M_k$ for any $k : 0 \leq k \leq n$. Firstly, we show *exactly to what extent* $\mathbb{E}[M_k]$ can be estimated from the sample $X^n$ and *how much remains* to be estimated from the underlying distribution $p$ and the number of elements $|\mathcal{X}|$. Specifically, we show the following.

**Theorem 1.1.**

$$\mathbb{E}[M_k] = \binom{n}{k} \left[ \sum_{i=1}^{n-k} (-1)^{i-1} f_{k+i}(n) \middle/ \binom{n}{k+i} \right] + R_{n,k} \tag{5}$$

*where $R_{n,k} = \binom{n}{k}(-1)^{n-k} f_{n+1}(n+1)$ is the remainder.*

This decomposition shows that the GT estimator is the *first term* of $\mathbb{E}[M_k]$ using the plug-in estimator $\Phi_1$ for $f_1(n)$. Hence, it gives the *exact bias* of the GT estimator in the multinomial setting (which would incorrectly be identified as *unbiased* using the Poisson approximation). We discuss bias and variance for the estimator $\hat{M}_k^B = \binom{n}{k} \left[ \sum_{i=1}^{n-k} (-1)^{i-1} \Phi_{k+i} \middle/ \binom{n}{k+i} \right]$ that is induced by Theorem 1.1.

Secondly, using our new theory, we cast the distribution-free estimation of $M_k$ as a search problem whose goal it is to find a distribution-specific estimator with a minimized MSE. Using the relationship in Eqn. 4 in Theorem 1.1, we notice many *representations* of $\mathbb{E}[M_k]$, all of which suggest different estimators for $\mathbb{E}[M_k]$. We introduce a deterministic method to construct a unique estimator from a representation, and show how to estimate the mean squared error (MSE) for such an estimator. Equipped with a large *search space* of representations and a *fitness function* to estimate the MSE of a candidate estimator, we can finally define our distribution-free estimation methodology.

We compare the performance of the minimal-bias estimator $\hat{M}_k^B$ and the minimal-MSE estimators discovered by our genetic algorithm to the that of the widely used GT estimator on a variety of multinomial distributions used for evaluation in previous work. Our results show that 1) the minimal-bias estimator has a substantially smaller bias than the GT estimator by thousands of order of magnitude, 2) Our genetic algorithm can produce estimators with MSE smaller than the GT estimator over 93% of the time when there are at least as many samples as classes; their MSE is roughly 80% of the GT estimator. We also publish all data and scripts to reproduce our results.

## 2 DEPENDENCIES BETWEEN FREQUENCIES $N_x$

We propose a new, distribution-free[1] methodology for reasoning about properties of estimators of the missing and total probability masses for multinomial distributions. The *main challenge* for the statistical analysis of $M_k$ has been reasoning in the presence of dependencies between frequencies $N_x$ for different elements $x \in \mathcal{X}$. As discussed in Section 1.1, a Poisson approximation with parameter $\lambda_x = p_x n$ is often used to render these frequencies as independent (Orlitsky & Suresh, 2015; Orlitsky et al., 2016; Acharya et al., 2013; Efron & Thisted, 1976; Valiant & Valiant, 2016; Good, 1953; Good & Toulmin, 1956; Hao & Li, 2020). In the following, we tackle this challenge by formalizing these dependencies between frequencies. Thus, we establish a link between the expected values of the corresponding total probability masses.

### 2.1 DEPENDENCY AMONG FREQUENCIES

Recall that the expected value $f_k(n)$ of the number of elements $\Phi_k$ that appear exactly $k$ times in the sample $X^n$ is defined as $f_k(n) = \sum_{x \in \mathcal{X}} \binom{n}{k} p_x^k (1-p_x)^{n-k}$. For convenience, let $g_k(n) = f_k(n)/\binom{n}{k}$. We notice the following relationship among $k$ and $n$:

$$g_k(n+1) = \sum_{x=1}^{S} p_x^k (1-p_x)^{n-k} \cdot (1-p_x) = g_k(n) - g_{k+1}(n+1) \tag{6}$$

$$= \sum_{i=0}^{n-k} (-1)^i g_{k+i}(n) + (-1)^{n-k+1} g_{n+1}(n+1) \tag{7}$$

---

[1] A *distribution-free analysis* is free of assumptions about the shape of the probability distribution generating the sample. In this case, we make no assumptions about parameters $p$ or $n$.

We can now write the expected value $\mathbb{E}[M_k]$ of the total probability mass in terms of the frequencies with which different elements $x \in \mathcal{X}$ have been observed in the sample $X^n$ of size $n$ as follows

$$\mathbb{E}[M_k] = \sum_{x \in \mathcal{X}} \binom{n}{k} p_x^{k+1} (1-p_x)^{n-k} = \binom{n}{k} g_{k+1}(n+1) = \binom{n}{k} \left[ \sum_{i=1}^{n-k} (-1)^{i-1} g_{k+i}(n) \right] + R_{n,k} \quad (8)$$

where $R_{n,k} = \binom{n}{k}(-1)^{n-k} f_{n+1}(n+1)$ is a remainder term. **This demonstrates Theorem 1.1.**

Figure 1 illustrates the relationship between the expected frequency of frequencies $f_k(n) = g_k(n)/\binom{n}{k}$, the frequency $k$, and the sample size $n$. The y- and x-axis represents the sample size $n$ and the frequency $k$, respectively. As per Eqn. (6), for every $2 \times 2$ lower triangle matrix, the value of the lower left cell ($g_k(n+1)$) is value of the upper left cell ($g_k(n)$) minus the value of the lower right cell ($g_{k+1}(n+1)$).

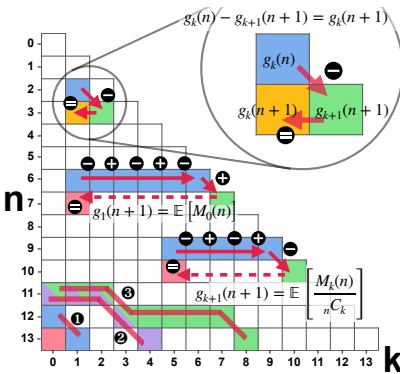

Figure 1: $g_k(n)$ lower triangle matrix

We can use this visualization to quickly see how to rewrite $g_k(n)$ as an alternating sum of values of the cells in the upper row, starting from the cell in the same column to the rightmost cell, and adding/subtracting the value of the rightmost cell in the current row. For instance, the $g_0(13)$ in the bottom-leftmost red cell in Figure 1 is equivalent to the various linear combinations of its surrounding cells: ❶ with $g_0(12)$ and $g_1(13)$ (blue colored), ❷ with $g_0(11)$, $g_1(11)$, $\cdots$, $g_4(13)$ (purple colored), or ❸ with $g_0(11)$, $g_1(11)$, $\cdots$, $g_8(13)$ (green colored).

**Missing Mass**. The missing probability mass $M_0$ gives the proportion of *all possible observations* for which the elements $x \in \mathcal{X}$ have *not* been observed in $X^n$. The expected value of $M_0$ is

$$\mathbb{E}[M_0] = g_1(n+1) = \left[ \sum_{k=1}^{n} (-1)^{k-1} g_k(n) \right] + (-1)^n f_{n+1}(n+1) \quad (9)$$

by Eqn. (8). The values in the second column of Figure 1 ($k=1$) represents the expected values of missing mass; $\mathbb{E}[M_0]$ being the cumulative sum of $(-1)^{k-1} g_k(n)$ is intuitively clear from the figure (the red cell in the row $n = 7$). It is here where we observe that $\mathbb{E}[M_0] = g_1(n+1)$ is *almost entirely determined* by the $g_*(n)$, the expected frequencies of frequencies in the sample $X^n$, and *not* by the number of elements $|\mathcal{X}|$ or their underlying distribution $p$. In fact, the influence of $p$ in the remainder term decays exponentially, i.e., $f_{n+1}(n+1) = \sum_{x \in \mathcal{X}} p_x^{n+1} \leq \sum_{x \in \mathcal{X}} \left( e^{1-p_x} \right)^{-n-1}$ which is dominated by the discovery probability of the most abundant element $\max(p)$.

**Total Mass**. Similarly, the expected value of the total probability mass $\mathbb{E}[M_k]$ (the red cell in the row $n = 10$), which is equal to $\binom{n}{k} g_{k+1}(n+1)$, is almost entirely determined by the expected frequencies of the sample $X^n$ with remainder $R_{n,k} = \binom{n}{k} \sum_{x \in \mathcal{X}} p_x^{n+1}$.

## 3    A LARGE CLASS OF ESTIMATORS

From the representation of $\mathbb{E}[M_k]$ in terms of frequencies in Eqn. (8) and the relationship across frequencies in Eqn. (6), we can see that there is a large number of representations of the expected total probability mass $\mathbb{E}[M_k]$. Each representation might suggest different estimators.

### 3.1    ESTIMATOR WITH EXPONENTIALLY DECAYING BIAS

We start by defining the minimal bias estimator $\hat{M}_k^B$ from the representation in Eqn. (8) and explore its properties. Let

$$\hat{M}_k^B = -\binom{n}{k} \sum_{i=1}^{n-k} \frac{(-1)^i \Phi_{k+i}}{\binom{n}{k+i}} \quad (10)$$

Table 1: $\mathbb{E}[M_k]$ preserving identites and example representations.

| $\mathbb{E}[M_k]$ preserving identites | Initial representation $r_0$ | Example representation $r_1$ |
| --- | --- | --- |
| $g_i(j) =_{(1)} ((1-\delta)g_i(j) + \delta g_i(j))$ 
 $=_{(2)} (g_i(j+1) + g_{i+1}(j+1))$ 
 $=_{(3)} (g_i(j-1) - g_{i+1}(j))$ 
 $=_{(4)} (g_{i-1}(j-1) + g_{i-1}(j))$ | $\alpha_{i,j} = \begin{cases} \binom{n}{k} & \text{for } i = k+1 \text{ and } j = n+1 \\ 0 & \text{otherwise.} \end{cases}$ | $\alpha_{i,j} = \begin{cases} \binom{n}{k}\big/2 & \text{for } i = k+1 \text{ and } j = n+1 \\ \binom{n}{k}\big/2 & \text{for } i = k+1 \text{ and } j = n \\ -\binom{n}{k}\big/2 & \text{for } i = k+2 \text{ and } j = n+1 \\ 0 & \text{otherwise.} \end{cases}$ |

**Bias**. For some constant $k : 0 \le k \le n$ and some constant $\mathbf{c} > 1$, the bias of $\hat{M}_k^B$ is in the order of $\mathcal{O}(n^k \mathbf{c}^{-n})$, i.e., $|Bias_B| = \left| \mathbb{E}\left[\hat{M}_k^B - M_k\right] \right| = R_{n,k} = \binom{n}{k} \sum_{x \in \mathcal{X}} p_x^{n+1} \le \binom{n}{k} \sum_{x \in \mathcal{X}} \mathbf{c}_x^{-n} \le n^k \sum_{x \in \mathcal{X}} \mathbf{c}_x^{-n}$, where $\mathbf{c}_x > 1$ for all $x \in \mathcal{X}$ are constants.

**Variance**. The variance of $\hat{M}_k^B$ is given by the variances and covariances of the $\Phi_{k+i}$ for $i \in [1..n-k]$. Under the certain conditions, the variance of $\hat{M}_k^B$ also decays exponentially in $n$.

**Theorem 3.1.** $\mathrm{Var}\left(\hat{M}_k^B\right)$ *decreases exponentially with* $n$ *if* $p_{\max} < 0.5$ *or* $\frac{(1-p_{\max})(1-p_{\min})}{p_{\max}} < 1$, *where* $p_{\max} = \max_{x \in \mathcal{X}} p_x$ *and* $p_{\min} = \min_{x \in \mathcal{X}} p_x$. *The proof is postponed to Appendix B.*

**Comparison to Good-Turing** (GT). The bias of $\hat{M}_k^B$ not only decays exponentially in $n$ but is also *smaller* than that of GT estimator $\hat{M}_k^G$ by an exponential factor. For a simpler variant of GT estimator, $\hat{M}_k^{G'} = \frac{k+1}{n-k} \Phi_{k+1}$ (suggested in McAllester & Schapire (2000)), which corresponds to the first term in the expected total probability mass $\mathbb{E}[M_k]$ in Eqn. (8), we show that its bias is *larger by an exponential factor* than the absolute bias of $\hat{M}_k^B$. To see this, we provide bounds on the individual sums and then on the bias ratio:

$$\mathbb{E}\left[\hat{M}_k^{G'} - M_k\right] \ge \binom{n}{k} p_{\min}^{k+2} (1-p_{\min})^{n-k-1} \quad (11)$$

$$\left| \mathbb{E}\left[\hat{M}_k^B - M_k\right] \right| \le \binom{n}{k} S p_{\max}^{n+1}, \quad (12)$$

$$\therefore \left| \frac{Bias_{G'}}{Bias_B} \right| \ge \frac{p_{\min}^{k+2}}{S p_{\max}^{k+2}} \left( \frac{1-p_{\min}}{p_{\max}} \right)^{n-k-1},$$

where $S = |\mathcal{X}|$. Noticing that $(1-p_{\min})/p_{\max} > 1$ for all distributions over $\mathcal{X}$, except where $S = 2$ and $p = \{0.5, 0.5\}$, the ratio decays exponentially in $n$ for $k \ll n$. The same can be shown for the original GT estimator $\hat{M}_k^G = \frac{k+1}{n} \Phi_{k+1}$ for a sufficiently large sample size (see Appendix A). For instance, the missing mass $M_0$ for the uniform distribution is overestimated by $\hat{M}_0^G$ on the average by $(S-1)^{n-1}/S^n$ while $\hat{M}_0^B$ has a bias of $(-1)^n/S^n$, which is lower by a factor of $1/(S-1)^{n-1}$.

While the bias of our estimator $\hat{M}_k^B$ is lower than that of $\hat{M}_0^G$ by an exponential factor, the variance is higher. The variance of $\hat{M}_k^B$ depends on the variances of and covariances between $\Phi_{k+i}$s:

$$\mathrm{Var}\left(\hat{M}_k^B\right) = \sum_{i=1}^{n-k} c_i^2 \mathrm{Var}(\Phi_{k+i}) + \sum_{i \ne j} (-1)^{i+j} c_i c_j \mathrm{Cov}(\Phi_{k+i}, \Phi_{k+j}), \quad (13)$$

where $c_i = \binom{n}{k} \big/ \binom{n}{k+i}$. In contrast, the variance of $\hat{M}_k^G$ depends only on the variance of $\Phi_{k+1}$. Later, we empirically investigate the bias and the variance of the two estimators.

## 3.2 Estimation with Minimal MSE as Search Problem

There are many representations of $\mathbb{E}[M_k] = \binom{n}{k} g_{k+1}(n+1)$ that can be constructed by recursively rewriting terms according to the dependency among frequencies we identified (cf. Eqn. (6 & 8)). The representation used to construct our minimal-bias estimator $\hat{M}_k^B$ was one of them. However, we notice that the variance of $\hat{M}_k^B$ is too high to be practical.

To find a representation from which an estimator with a minimal mean squared error (MSE) can be derived, we cast the estimation of $M_k$ as an *optimization problem*. We first define the *search space* of representations of $\mathbb{E}[M_k]$ and the *fitness function* to estimate the MSE of a candidate estimator.

**Search space**. Let $\mathbb{E}[M_k]$ be *represented* by a suitable choice of coefficients $\{\alpha_{i,j}\}$ such that $\mathbb{E}[M_k] = \sum_{i=1}^{n+1} \sum_{j=i}^{n+1} \alpha_{i,j} g_i(j)$; the search space of our optimization problem is the set of all possible representations of $\mathbb{E}[M_k]$. One representation of $\mathbb{E}[M_k] = \binom{n}{k} g_{k+1}(n+1)$ is $r_0$ in Table 1.

Different encodings can be used to express the search space. In a Markov chain encoding, we realize that the four identities (1-4) in the first column of Table 1 can be recursively applied to the initial representation $r_0$ in the second column of Table 1 to explore the search space. For instance, applying identity (1) with $\delta = 0.5$ and identity (3) to $r_0$, we obtain the representation $r_1$ in Table 1. In a continuous optimization problem encoding, we realize that the coefficients $\alpha_{i,j}$ must satisfy the following constraints: for $\forall 1 \leq k' \leq n + 1$,

$$\sum_{i=1}^{n+1} \sum_{j=i}^{n+1} c_{i,j} \alpha_{i,j} = \begin{cases} \binom{n}{k} & \text{if } k' = k+1, \\ 0 & \text{otherwise,} \end{cases} \quad \text{where } c_{i,j} = \begin{cases} \binom{n+1-j}{k'-i} & \text{if } 0 \leq k'-i \leq n+1-j, \\ 0 & \text{otherwise,} \end{cases} \quad (14)$$

and any choice of $\alpha_{i,j}$ that satisfy Eqn. (14) is in the search space. Appendix C provides proof of how each encoding expresses the search space.

**Estimator instantiation**. To construct a unique estimator $\hat{M}_k^r$ of $M_k$ from a representation $r$ of $\mathbb{E}[M_k]$, we propose a deterministic method. But first, we define our random variables on subsamples of $X^n$. For any $m \leq n$, let $N_x(m)$ be the number of times element $x \in \mathcal{X}$ is observed in the subsample $X^m = \langle X_1, \cdots X_m \rangle$ of $X^n$. Let $\Phi_k(m)$ be the number of elements appearing exactly $k$ times in $X^m$, i.e., $N_x(m) = \sum_{i=1}^{m} \mathbf{1}(X_i = x)$, $\Phi_k(m) = \sum_{x \in \mathcal{X}} \mathbf{1}(N_x(m) = k)$. Note that $N_x = N_x(n)$ and $\Phi_k = \Phi_k(n)$. Hence, given a representation $r$, we can construct $\hat{M}_k^r$ as

$$\hat{M}_k^r = \left[ \sum_{i=1}^{n} \sum_{j=i}^{n} \frac{\alpha_{i,j}}{\binom{j}{i}} \Phi_i(j) \right] + \left[ \sum_{i=1}^{n} \frac{\alpha_{i,n+1}}{\binom{n+1}{i}} \Phi_i \right] \quad (15)$$

Notice that $\Phi_i(j) \big/ \binom{j}{i}$ is just the plug-in estimator for $g_i(j)$.

**Fitness function**. To define the quantity to optimize, any optimization problem requires a fitness (objective) function. Our *fitness function* takes a candidate representation $r$ and returns an estimate of the MSE of the corresponding estimator $\hat{M}_k^r$. We decompose the MSE as the sum of the squared bias, variances of, and the covariance between $M_k$ and $\hat{M}_k^r$. For convenience, let $f_{n+1}(n) = 0$.

$$\text{MSE}(\hat{M}_k^r) = \left[ \sum_{1 \leq i \leq j \leq n} \beta_{i,j}^2 f_i(j) - \binom{n}{k} g_{k+1}(n+1) \right]^2 + \sum_{\substack{1 \leq i \leq j \leq n \\ 1 \leq l \leq m \leq n}} \beta_{i,j} \beta_{l,m} \text{Cov}\left( \Phi_i(j), \Phi_l(m) \right)$$

$$+ \text{Var}\left( M_k \right) - 2 \sum_{1 \leq i \leq j \leq n} \beta_{i,j} \text{Cov}\left( \Phi_i(j), M_k \right), \quad (16)$$

where $\beta_{i,j}$ is the coefficient of $\Phi_i(j)$ in $\hat{M}_k^r$. We expand on the MSE computation in Appendix D. The resulting estimator $\hat{M}_k^r$ minimizing the fitness function provides a estimator with a minimal MSE regarding the arbitrary sample of size $n$ from the underlying distribution.

Since the underlying distribution $\{p_x\}_{x \in \mathcal{X}}$ is *unknown*, we can only *estimate* the MSE. For any element $x$ that has been observed exactly $k > 0$ time in the sample $X^n$, we use $\hat{p}_x = \hat{M}_k^G / \Phi_k$ as natural estimator of $p_x$, where $\hat{M}_k^G$ is the GT estimator. To handle unobserved elements ($k = 0$), we first estimate the number of unseen elements $\mathbb{E}[\Phi_0] = f_0(n)$ using Chao's nonparamteric species richness estimator $\hat{f}_0 = \frac{n-1}{n} \frac{\Phi_1^2}{2\Phi_2}$ (Chao, 1984), and then estimate the probability of each such unseen element as $\hat{p}_y = \hat{M}_0^G / \hat{f}_0$, where $\hat{M}_0^G$ is the GT estimator. Finally, we plug these estimates into Eqn. (16) to estimate the MSE. It is interesting to note that it is precisely the GT estimator whose MSE our approach is supposed to improve upon.

**Optimization algorithm**. With the required concepts in place, different optimization algorithms can be used to find the representation of $\mathbb{E}[M_k]$ that minimizes the MSE of the estimator. The performance guarantee for the discovered estimator inherently depends on the optimization algorithm.

We develop a genetic algorithm (GA) (Mitchell, 1998) for the optimization problem encoded with the Markov chain encoding. Appendix E provides the GA in detail. Here, we briefly sketch the general procedure. Starting from the *initial representation* $r_0$ in Table 1, our GA iteratively improves a population of candidate representations $P_g$, called *individuals*. For every generation $g$, our GA selects the $m$ fittest individuals from the previous generation $P_{g-1}$, mutates them by randomly applying the $\mathbb{E}[M_k]$ preserving identites (1-4) in Table 1, and creates the current generation $P_g$ by

adding the Top-$n$ individuals from the previous generation, i.e., the *elitism* strategy that guarantees that the best individuals are not lost. The GA repeats this process for the iteration limit $G_L$ and outputs the best evolved estimator $\hat{M}_k^{\text{Evo}}$. Later, we empirically evaluate the performance of the discovered estimator $\hat{M}_k^{\text{Evo}}$ against the GT estimator $\hat{M}_k^G$.

Another approach one could use is a quadratic programming (Nocedal & Wright, 2006) on the continuous optimization problem encoding. The fitness function, Eqn (16), is quadratic in the coefficients $\beta_{i,j}$, and the constraints, Eqn (14), are linear. Also, the search space is convex as the quadratic term in the fitness function (i.e., the sum of the variance and the square of the expected value of the estimator) is non-negative. By Ye & Tse (1989), quadratic programming finds the optimal solution in a number of iterations that is polynomial in the sample size $n$. However, we notice that the fitness evaluation is prohibitively high, in the order of $\mathcal{O}(n^4)$ due to the number of covariance terms in Eqn (16), which makes using quadratic programming less practical for large $n$. Unlike quadratic programming, we found our GA is more practical for large $n$ as only the variance and covariance terms that are needed to compute the fitness function are computed.

**Adapting to a larger sample**. The estimator $\hat{M}_k^{\text{Evo}}$ discovered by the optimization for samples of size $n$ can be converted to an estimator for a sample of arbitrary size $m \geq n$. Notice that the representation of $\mathbb{E}[M_k]$ of the sample of size $n$ can be converted to a representation of $\mathbb{E}[M_k']$ of a larger sample of size $m$ by multiplying the coefficients $\alpha_{i,j}$ by $\binom{m}{k}/\binom{n}{k}$. Especially for the missing mass ($k = 0$), the representation is valid for any sample size $m \geq n$ without modification. This property can be used to easily derive the missing mass estimator for a larger sample size $m > n$. Given the coefficients $\alpha_{i,j}$ discovered by the optimization for the missing mass of a sample $X^n \sim p$, the estimator of the missing mass for any sample $X^m \sim p$ is given as

$$\left[\sum_{i=1}^{n}\sum_{j=i+m-n}^{m} \frac{\alpha_{i,j+n-m}}{\binom{j}{i}}\Phi_i(j)\right] + \left[\sum_{i=1}^{n} \frac{\alpha_{i,n+1}}{\binom{n+1}{i}}\Phi_i\right]. \tag{17}$$

It is worth noting that the adapted estimator from the minimal-MSE estimator for a sample of size $n$ to a sample of size $m$ is not necessarily the minimal-MSE estimator for the sample of size $m$. Yet, it lessens the computational burden of finding the minimal-MSE estimator for a larger sample size and may have a lower MSE (than the GT estimator) if the (relative variance of the) frequencies $\Phi$ are similar in the extended sample. We empirically investigate this property in our experiments.

**Distribution-free**. While our approach itself is *distribution-free*, the output is *distribution-specific*, i.e., the discovered estimator has a minimal MSE on the specific, unknown distribution.

## 4 EXPERIMENT

We design experiments to evaluate the performance (i) of our minimal-bias estimator $\hat{M}_k^B$ and (ii) of our the minimal-MSE estimator $\hat{M}_k^{\text{Evo}}$ that is discovered by our genetic algorithm against the performance of the widely-used Good-Turing estimator $\hat{M}_k^G$ (Good, 1953).

**Distibutions**. We use the same six multinomial distributions that are used in previous evaluations (Orlitsky & Suresh, 2015; Orlitsky et al., 2016; Hao & Li, 2020): a uniform distribution (uniform), a half-and-half distribution where half of the elements have three times of the probability of the other half (half&half), two Zipf distributions with parameters $s = 1$ and $s = 0.5$ (zipf-1, zipf-0.5), and distributions generated by Dirichlet-1 prior and Dirichlet-0.5 prior (diri-1, diri-0.5, respectively).

**Open Science and Replication**. For scrutiny and replicability, we publish all our evaluation scripts at: `https://github.com/niMgnoeSeeL/UnseenGA`.

### 4.1 EVALUATING OUR MINIMAL-BIAS ESTIMATOR

- **RQ1**. *How does our estimator for the missing mass $\hat{M}_0^B$ compare to the Good-Turing estimator $\hat{M}_0^G$ in terms of bias as a function of sample size $n$?*
- **RQ2**. *How does our estimator for the total mass $\hat{M}_k^B$ compare to the Good-Turing estimator $\hat{M}_k^G$ in terms of bias as a function of frequency $k$?*
- **RQ3**. *How do the estimators compare in terms of variance and mean-squared error?*

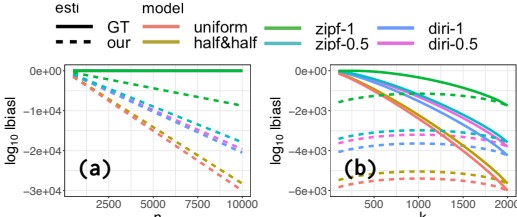

Figure 2: Absolute bias of $\hat{M}_0^B$ and $\hat{M}_0^G$ (a) as a function of $n$ for $k = 0$ and (b) as a function of $k$ for $n = 2000$ ($S = 1000$, log-scale).

| $n$ | $\hat{M}_0$ | Bias | Var | MSE |
|---|---|---|---|---|
| 100 | GT | 3.6973e-003 | 2.3372e-03 | 2.3508e-03 |
| | ours | 1.0000e-200 | 2.3515e-03 | 2.3515e-03 |
| 500 | GT | 6.6369e-005 | 1.1430e-05 | 1.1434e-05 |
| | ours | $<$ 1.00e-200 | 1.1445e-05 | 1.1445e-05 |
| 1000 | GT | 4.3607e-007 | 4.3439e-08 | 4.3439e-08 |
| | ours | $<$ 1.00e-200 | 4.3441e-08 | 4.3441e-08 |

Table 2: Bias, variance, and MSE of $\hat{M}_0^B$ and $\hat{M}_0^G$ for three values of $n$. (uniform, $S = 100$). More results are in Appendix F.

We focus specifically on the *bias* of $\hat{M}_k^B$, i.e., the average difference between the estimate and the expected value $\mathbb{E}[M_k]$. We expect that the bias of the *missing mass* estimate $\hat{M}_0^B$ as a function of $n$ across different distributions provides empirical insight for our claim that how much is unseen chiefly depends on information about the seen.

**RQ.1**. Figure 2(a) illustrates how fast our estimator $\hat{M}_k^B$ and the baseline estimator $\hat{M}_k^G$ (GT) approach the expected missing mass $\mathbb{E}[M_0]$ as a function of sample size $n$. As it might difficult for the reader to discern differences across distributions for the baseline estimator, we refer to Appendix F, where we zoom into a relevant region. The *magnitude* of our estimator's bias is significantly smaller than the magnitude of GT's bias *for all distributions* (by thousands of orders of magnitude). Figure 2(a) also nicely illustrates the *exponential decay* of our estimator in terms of $n$ and how our estimator is less biased than GT by an exponential factor.

In terms of distributions, a closer look at the performance differences confirms our suspicion that the bias of our estimator is strongly influenced by the probability $p_{\max}$ of the most abundant element. In fact, by Eqn. (12) the absolute bias of our estimator is minimized when $p_{\max}$ is minimized. If we ranked the distributions by values of $p_{\max}$ with the smallest value first ⟨uniform, half&half, zipf-0.5, zipf-1⟩,[2] we would arrive at the same ordering in terms of performance of our estimator as shown in Figure 2(a). Similar observations can be made for GT, but the ordering is different (check Appendix F).

**RQ2**. Figure 2(c) illustrates for both estimators of the *total mass* $M_k$ how the bias behaves as $k$ varies between 0 and $n = 2000$ when $S = 1000$. The trend is clear; the bias of our estimator is strictly smaller than the bias of GT for all $k$ and all the distributions. The difference is the most significant for rare elements (small $k$) and gets smaller as $k$ increases. The bias of our estimator is maximized when $k = 1000 = 0.5n$, the bias for GT when $k = 0$.

**RQ3**. Table 2 shows variance and MSE of both estimators for the missing mass $M_0$ for the uniform and three values of $n$. As we can see, the MSE of our estimator is approximately the same as that of GT. The reason is that the MSE is dominated by the variance. We make the same observation for all other distributions (see Appendix F). The MSEs of both estimators are comparable.

## 4.2 EVALUATING OUR ESTIMATOR DISCOVERY ALGORITHM

- **RQ1** (Effectiveness). *How does our estimator for the missing mass $\hat{M}_0^{Evo}$ compare to the Good-Turing estimator $\hat{M}_0^G$ in terms of MSE?*[3]

- **RQ2** (Efficiency). *How long does it take for our GA to generate an estimator $M_k^{Evo}$?*

- **RQ3** (Larger Sample). *How does $\hat{M}_k^{Evo}$ generated from a sample of size $n$ perform on a sample of size $m > n$?*

- **RQ4** (Distribution-awareness). *How well does an estimator discovered from a sample from one distribution perform on another distribution in terms of MSE?*

- **RQ5** (Empirical Application) *How does our estimator perform in a real-world application?*

---

[2]diri-1 and diri-0.5 are not considered because multiple distributions are sampled from the Dirichlet prior.

[3]We also considered more recent related work that can be considered to estimate the missing mass (Painsky, 2023; Valiant & Valiant, 2017; Wu & Yang, 2019); the results are in Appendix G.

Table 3: The MSE of the best evolved estimator $M_0^{\text{Evo}}$ and GT estimator $\hat{M}_0^G$ for the missing mass $M_0$, the success rate $\hat{A}_{12}$, and the ratio (Ratio, $MSE(\hat{M}_0^{\text{Evo}})/MSE(M_0^G)$) for three sample sizes $n$ and six distributions with support size $S = 100$.

| Dist. | $n = S/2$ | | | | $n = S$ | | | | $n = 2S$ | | | |
|---|---|---|---|---|---|---|---|---|---|---|---|---|
| | $MSE(\hat{M}_0^G)$ | $MSE(M_0^{\text{Evo}})$ | $\hat{A}_{12}$ | Ratio | $MSE(\hat{M}_0^G)$ | $MSE(M_0^{\text{Evo}})$ | $\hat{A}_{12}$ | Ratio | $MSE(\hat{M}_0^G)$ | $MSE(M_0^{\text{Evo}})$ | $\hat{A}_{12}$ | Ratio |
| uniform | 1.09e-02 | 7.94e-03 | 0.88 | 72% | 6.05e-03 | 4.29e-03 | 0.97 | 70% | 1.93e-03 | 1.73e-03 | 0.96 | 89% |
| half&half | 1.14e-02 | 7.16e-03 | 0.90 | 63% | 5.46e-03 | 4.07e-03 | 0.98 | 74% | 1.57e-03 | 1.42e-03 | 0.93 | 90% |
| zipf-1 | 8.09e-03 | 7.37e-03 | 0.87 | 91% | 3.42e-03 | 3.04e-03 | 0.89 | 88% | 1.26e-03 | 1.08e-03 | 0.94 | 85% |
| zipf-0.5 | 1.08e-02 | 8.13e-03 | 0.91 | 75% | 5.23e-03 | 4.16e-03 | 0.96 | 79% | 1.73e-03 | 1.54e-03 | 0.97 | 88% |
| diri-1 | 1.10e-02 | 7.97e-03 | 0.92 | 72% | 4.36e-03 | 3.47e-03 | 0.92 | 79% | 1.23e-03 | 1.05e-03 | 0.91 | 85% |
| diri-0.5 | 9.90e-03 | 8.02e-03 | 0.87 | 81% | 3.47e-03 | 2.86e-03 | 0.88 | 82% | 9.41e-04 | 8.08e-04 | 0.86 | 85% |
| Avg. | | | 0.89 | 76% | | | 0.93 | 79% | | | 0.93 | 87% |

| Dist. | $c = 2$ | | $c = 5$ | | $c = 10$ | |
|---|---|---|---|---|---|---|
| | Ratio | $p < .05$ | Ratio | $p < .05$ | Ratio | $p < .05$ |
| uniform | 1.00 | False | 1.00 | False | 0.99 | True |
| half&half | 0.95 | True | 0.96 | True | 0.98 | True |
| zipf-0.5 | 0.97 | True | 0.98 | True | 0.99 | True |
| zipf-1 | 0.93 | True | 0.95 | True | 0.97 | True |
| diri-1 | 0.91 | True | 0.93 | True | 0.95 | True |
| diri-0.5 | 0.93 | True | 0.95 | True | 0.96 | True |

Table 4: The MSE comparison for the missing mass $M_0$ ($S = 100$, $n = 100$) for extended samples $X^{cn}$ ($c \in \{2, 5, 10\}$) between the GT estimator $\hat{M}_0^G$ and the adapted estimator from the evolved estimator $\hat{M}_0^{\text{Evo}}$ for $X^n$. 'Ratio' is the ratio of the MSE ($MSE(\hat{M}_0^{\text{Evo}})/MSE(M_0^G)$) and '$p < .05$' is the result of the (one-sided) Wilcoxon signed-rank test.

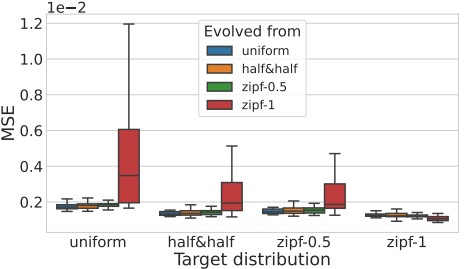

Figure 3: The MSE of an estimator discovered using a sample ($S, n = 100, 200$) from one distribution (individual boxes) applied to another target distribution (box clusters).

To handle the randomness in our evaluation, we repeat each experiment we repeat the experiments 100 times, i.e., we take 100 different samples $X^n$ of size $n$. More details about our experimental setup can be found in Appendix E.

**RQ.1** (Effectiveness). Table 3 shows average MSE of the estimator $M_0^{\text{Evo}}$ discovered by our genetic algorithm and that of the GT estimator $\hat{M}_0^G$ for the missing mass $M_0$ across three sample sizes. We measure effect size using *Vargha-Delaney* $\hat{A}_{12}$ (Vargha & Delaney, 2000) (*success rate*), i.e., the probability that the MSE of the estimator discovered by our genetic algorithm has a smaller MSE than the GT estimator (*larger is better*). Moreover, we measure the MSE of our estimator as a proprtion of the MSE of GT, called *ratio* (*smaller is better*). Results for other $S$ is in Appendix F.

Overall, the estimator discovered by our GA performs *significantly* better than GT estimator in terms of MSE (avg. $\hat{A}_{12} > 0.89$; $ratio < 87\%$). The performance difference increases with sample size $n$. When the sample size is twice the support size ($n = 2S$), in 93% of runs our discovered estimator performs better. The average MSE of our estimator is somewhere between 76% and 87% of the MSE of GT. The high success rate and the low ratio of the MSE shows that the GA is effective in finding the estimator with the minimal MSE for the missing mass $M_0$. A Wilcoxon signed-rank test shows that all performance differences are statistically significant at $\alpha < 10^{-9}$. In terms of distributions, the performance of our estimator is similar across all distributions, showing the generality of our algorithm. The worst performance is for the zipf-1 distribution, though it is still 85-91% of the GT estimator's MSE. The potential reason for this is due to the overfitting to the approximated distribution $\hat{p}_x$. Since the zipf-1 is the most skewed distribution, there are more elements unseen in the sample than in other distributions, which makes the approximated distribution $\hat{p}_x$ less accurate.

**RQ.2** (Efficiency). The time GA takes is reasonable; to compute an estimator in Table 3, it takes 57.2s on average (median: 45.3s). The average time per iteration is 0.19s (median: 0.16s).

**RQ.3** (Larger Sample). Table 4 shows how the estimator $\hat{M}_0^{\text{Evo}}$ that is discovered for a given sample $X^n$ of size $n$ performs on an extended larger sample $X^{cn}$ ($c \in \{2, 5, 10\}$) by adapting the coefficients $\alpha_{i,j}$ for the larger sample as described in Eqn. (17). To evaluate the performance, we sample $X^{cn-n}$ additional samples from the same distribution and compute the missing mass $M_0$ for the extended sample $X^{cn}$ using the adapted estimator as well as the GT estimator $\hat{M}_0^G$; the entire process is repeated 10K times to calculate the MSE. The results show that the adapted estimator performs

better than the GT estimator for most distributions and sample sizes. The ratio of the MSE is between 0.91 and 1.00, indicating that the adapted estimator performs better than the GT estimator but not as much as the evolved estimator for the original sample $X^n$, as they are not genuinely designed for the larger sample. The performance of the adapted estimator degraded for the uniform distribution. The possible reason is that because uniform has, technically, no *rare* classes, it is easier to find the unseen/rarely seen classes in the extended sample, making the relative variance between frequencies of frequencies ($\Phi$) differ a lot from the original sample. For instance, when $n = 100$, the variance of the number of singletons $\mathrm{Var}\,(\Phi_1(100))\,(\approx 23.37)$ is much larger than the variance of the number of doubletons $\mathrm{Var}\,(\Phi_2(100))\,(\approx 11.63)$ for the uniform distribution. However, the order already reverses when $n = 200$ ($\mathrm{Var}\,(\Phi_1(200)) \approx 16.04 < \mathrm{Var}\,(\Phi_2(200)) \approx 19.84$), which becomes more significant when $n = 1000$ ($\mathrm{Var}\,(\Phi_1(1000)) \approx 0.043 \ll \mathrm{Var}\,(\Phi_2(2000)) \approx 0.216$).

**RQ.4** (Distribution-awareness). Figure 3 shows the performance of an estimator discovered from a sample from one distribution (source) when applied to another distribution (target). Applying an estimate from the zipf-1 on the zipf-1 gives the optimal MSE (right-most red box). However, applying an estimator from the zipf-1 on the uniform (left red box) yields a huge increase in variance. In terms of effect size, we measure a Vargha Delaney $\hat{A}_{12} > 0.95$ between the "home" and "away" estimator. While the uniform also shows that the home estimator performs best on the home distribution ($\hat{A}_{12} = 0.61$ (small)), the difference between the estimators from uniform, half&half, and zipf-0.5 is less significant. Perhaps unsurprisingly, an estimator performs optimal when the source of the samples is similar to the target distribution.

**RQ.5** (Empirical Application). We briefly demonstrate the performance of our estimator in two real-world applications. We first use the Australian population-data-by-region dataset ((Arvidsson, 2023), $S = 104$) to estimate $M_0$ for $n = 50$ random data points. The ground truth $M_0$ is 0.476. The GT estimator $\hat{M}_0^G$ estimates $M_0$ as [0.322, 0.638] with a 95% confidence interval (CI) demonstrating a huge variance. In contrast, our estimator $\hat{M}_0^{\mathrm{Evo}}$ estimates $M_0$ as [0.409, 0.565] (95%-CI) demonstrating only 25% of the MSE of GT. We also apply our method to the Shakespeare dataset (Sha) ($S = 935$, $|\mathrm{Datstset}| = 111,396$), commonly used in the literature (Efron & Thisted, 1976), focusing on missing mass for player frequency. For $n = \langle 100, 200, 500 \rangle$, the MSE of our estimator is $\langle 3.0{\times}10^{-3}, 2.1{\times}10^{-3}, 8.8{\times}10^{-4} \rangle$ compared to the GT estimator with $\langle 4.2{\times}10^{-3}, 2.6{\times}10^{-3}, 9.0{\times}10^{-4} \rangle$, respectively, showing that our method consistently outperforms the Good-Turing estimator across all sample sizes. The result shows that our approach lead to a substantial and significant decrease of the MSE in the real-world application.

**Summary**. To summarize, our GA is effective in finding the estimator with the minimal MSE for the missing mass $M_0$ with the smaller MSE than GT estimator $\hat{M}_0^G$ for all distributions and sample sizes. The effect is substantial and significant and the average decrease of the MSE is roughly one fifth against GT estimator $\hat{M}_0^G$. We report results of additional experimental results, including the variance of the GA and probability mass estimation ($k > 0$), in Appendix F & G.

## 5  DISCUSSION

**Beyond the General Estimator.** In this study, we propose a "meta" estimation methodology that can be applied to a set of samples from a specific unknown distribution. The conventional approach is to develop *an estimator* for an arbitrary distribution. Yet, each distribution has its own characteristics, and, because of that, the "shape" of the (frequencies of) frequencies of the classes in the sample differs across distributions (e.g., between the uniform and the Zipf distribution). In contrast to the conventional approach, we propose a *distribution-free* methodology to discover the a *distribution-specific* estimator with low MSE (given only the sample). Note that, while we use the genetic algorithm to discover the estimator, any optimization method can be used to discover the estimator, for instance, a constrained optimization solver.

**Extrapolation.** Estimating the probability to discover a new species if the sample was enlarged by *one* is a well-known problem in many scientific fields, such as ecology, linguistics, and machine learning, and software testing (Lee et al., 2025; Liyanage et al.; Lee & Böhme, 2023; Böhme, 2022; 2018; Böhme et al., 2021). Given a sample of size $n$, what is the expected number $\mathbb{E}\,[U(t)]$ of new species discovered if $t$ times more samples were taken ($\mathbb{E}\,[U(t)] = f_0(n) - f_0(n + nt)$)? Good & Toulmin (1956) proposed a seminal estimator based on $\Phi_k$. Using the Poisson approximation, this estimator has been improved in several ways (Hao & Li, 2020). We believe that our analysis can be extended to the Good-Toulmin estimator seeking more accurate estimators for $U(t)$.

ACKNOWLEDGMENTS

We thank the anonymous reviewers for their valuable feedback. This work is partially funded by the European Union. The views and opinions expressed are however those of the author(s) only and do not necessarily reflect those of the European Union or the European Research Council Executive Agency. Neither the European Union nor the granting authority can be held responsible for them. This work is supported by ERC grant (Project AT SCALE, 101179366). This work is also partially funded by the Deutsche Forschungsgemeinschaft (DFG, German Research Foundation) under Germany's Excellence strategy - EXC2092 - Project #390781972.

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

## A   COMPARING THE BIAS OF THE ESTIMATORS

In Section 3.1, we have shown that the bias of a simpler variant of GT, $\hat{M}_k^{G'} = \frac{k+1}{n-k}\Phi_{k+1}$, is larger by an exponential factor than the absolute bias of our minimal bias estimator $\hat{M}_k^B$. In this section, we show that the bias of the original GT estimator $\hat{M}_k^G = \frac{k+1}{n}\Phi_{k+1}$ is also larger by an exponential factor than the absolute bias of $\hat{M}_k^B$ for a sufficiently larger sample size. Recall that

$$Bias_{G'} = \mathbb{E}\left[\hat{M}_k^G - M_k\right] = \frac{k+1}{n-k}f_{k+1}(n) - \binom{n}{k}g_{k+1}(n+1) = \sum_x \binom{n}{k}p_x^{k+2}(1-p_x)^{n-k-1}, \quad (18)$$

and

$$Bias_G = \mathbb{E}\left[\hat{M}_k^G - M_k\right] = \binom{n}{k}g_{k+2}(n+1) - \frac{k(k+1)}{n(n-k)}f_{k+1}(n) \tag{19}$$

$$= \binom{n}{k}g_{k+2}(n+1) - \binom{n-1}{k-1}g_{k+1}(n) \tag{20}$$

$$= \sum_x p_x^{k+2}(1-p_x)^{n-k-1}\left(\binom{n}{k} - \frac{1}{p_x}\cdot\binom{n-1}{k-1}\right) \tag{21}$$

$$= \sum_x \binom{n}{k}p_x^{k+2}(1-p_x)^{n-k-1}\left(1 - \frac{k}{n\cdot p_x}\right) \tag{22}$$

$$\geq \left(1 - \frac{k}{n\cdot p_{\max}}\right)Bias_{G'}, \tag{23}$$

where $1 - \frac{k}{n\cdot p_{min}} > 0$ when $n$ is sufficiently large. Above inequality leads to the following:

$$\frac{Bias_G}{Bias_{G'}} \geq \left(1 - \frac{k}{n\cdot p_{\max}}\right), \quad \text{while} \quad \frac{Bias_B}{Bias_{G'}} \leq \frac{Sp_{\max}^{k+2}}{p_{\min}^{k+2}}\left(\frac{1-p_{\min}}{p_{\max}}\right)^{-n+k+1}, \tag{24}$$

which proves our claim.

## B   BOUNDING THE VARIANCE OF $\hat{M}_k^B$

The variance of the linear combination of random variables is given by

$$\mathrm{Var}\left(\sum_i c_i X_i\right) = \sum_i c_i^2 \mathrm{Var}(X_i) + \sum_{i\neq j} c_i c_j \mathrm{Cov}(X_i, X_j). \tag{25}$$

Therefore, the variance and the covariance of $\Phi_i(n)$s are the missing pieces to compute the variance of $\hat{M}_k^B$.

**Theorem B.1.** *Given the multinomial distribution $p = (p_1, \ldots, p_S)$ with support size $S$, the variance of $\Phi_i = \Phi_i(n)$ from $n$ samples $X^n$ is given by*

$$\mathrm{Var}(\Phi_i(n)) = \begin{cases} f_i(n) - f_i(n)^2 + \sum_{x\neq y}\frac{n!}{i!^2(n-2i)!}p_x^i p_y^i(1-p_x-p_y)^{n-2i} & \text{if } 2i \leq n, \\ f_i(n) - f_i(n)^2 & \text{otherwise.} \end{cases} \tag{26}$$

*Proof.*

$$\text{Var}(\Phi_i) = \mathbb{E}\left[\Phi_i^2\right] - \mathbb{E}\left[\Phi_i\right]^2 \tag{27}$$

$$\mathbb{E}\left[\Phi_i^2\right] = \mathbb{E}\left[\left(\sum_x \mathbf{1}(N_x = i)\right)^2\right] \tag{28}$$

$$= \mathbb{E}\left[\sum_x \mathbf{1}(N_x = i) + \sum_{x \neq y} \mathbf{1}(N_x = i \wedge N_y = i)\right] \tag{29}$$

$$= \begin{cases} f_i(n) + \sum_{x \neq y} \frac{n!}{i!^2(n-2i)!} p_x^i p_y^i (1 - p_x - p_y)^{n-2i} & \text{if } 2i \leq n, \\ f_i(n) & \text{otherwise.} \end{cases} \tag{30}$$

$$\therefore \text{Var}(\Phi_i) = \begin{cases} f_i(n) + \sum_{x \neq y} \frac{n!}{i!^2(n-2i)!} p_x^i p_y^i (1 - p_x - p_y)^{n-2i} - f_i(n)^2 & \text{if } 2i \leq n, \\ f_i(n) - f_i(n)^2 & \text{otherwise.} \end{cases} \tag{31}$$

$\square$

Now we compute the upper bound of the variance of $\hat{M}_k^B$.

**Lemma B.2.**

$$\text{Var}(\Phi_i) \begin{cases} \leq S f_i(n) - f_i(n)^2 & \text{if } 2i \leq n. \\ = f_i(n) - f_i(n)^2 & \text{otherwise.} \end{cases} \tag{32}$$

*Proof.* From Theorem B.1,

$$\mathbb{E}\left[\Phi_i^2\right] = f_i(n) + \mathbb{E}\left[\sum_{x \neq y} \mathbf{1}(N_x = i \wedge N_y = i)\right] \tag{33}$$

$$\leq f_i(n) + (S - 1)\mathbb{E}\left[\sum_x \mathbf{1}(N_x = i)\right] \tag{34}$$

$$= S f_i(n) \quad (\text{if } 2i \leq n). \tag{35}$$

$$\tag{36}$$

The lemma directly follows from the above inequality. $\square$

**Lemma B.3.**

$$g_i(n) \leq S \cdot \beta_{\min}^{-n} o_{\max}^i,$$

*where $S = |\mathcal{X}|$, $p_{\max} = \max_{x \in \mathcal{X}} p_x$, $\beta_{\min} = \frac{1}{1 - p_{\min}}$, and $o_{\max} = \frac{p_{\max}}{1 - p_{\max}}$.*

*Proof.* $\frac{1}{1-x}$ and $\frac{x}{1-x}$ are increasing functions for $x \in (0, 1)$. Therefore,

$$g_i(n) = \sum_{x \in \mathcal{X}} p_x^i (1 - p_x)^{n-i} = \sum_{x \in \mathcal{X}} \left(\frac{1}{1 - p_x}\right)^{-n} \left(\frac{p_x}{1 - p_x}\right)^i \leq |\mathcal{X}| \cdot \beta_{\min}^{-n} o_{\max}^i.$$

$\square$

**Theorem B.4.** *The variance of the estimator $\hat{M}_k^B$ is bounded as follows:*

$$\text{Var}(\hat{M}_k^B) \leq c_1 \cdot n^{2k+1} \cdot c_2^{-n},$$

*where $c_1 = S \cdot \left(\frac{e}{k}\right)^{2k}$, $c_2 = \min\left(\frac{1}{1-p_{\min}}, \frac{1-p_{\max}}{p_{\max}(1-p_{\min})}\right)$. In other words, $\text{Var}(\hat{M}_k^B) = \mathcal{O}(n^{2k+1} \cdot \beta_{\min}^{-n} \cdot \max(1, o_{\max}^n)).$*

*Proof.* From Lemma B.2 in the supplementary, we have $\text{Var}(\Phi_i) \leq S f_i(n) - f_i(n)^2$. Thus,

$$\text{Var}\left(\frac{\Phi_{k+i}}{\binom{n}{k+i}}\right) \leq \frac{S f_{k+i}(n) - f_{k+i}(n)^2}{\binom{n}{k+i}^2} \leq S \cdot \frac{g_{k+i}(n)}{\binom{n}{k+i}} \leq \frac{S^2 \cdot \beta_{\min}^{-n} o_{\max}^{k+i}}{\binom{n}{k+1}} \quad \text{(by Lemma B.3)} \tag{37}$$

$$\binom{n}{k}^2 \text{Var}\left(\frac{\Phi_{k+i}}{\binom{n}{k+i}}\right) \leq S\beta_{\min}^{-n} \frac{\binom{n}{k}^2}{\binom{n}{k+1}} o_{\max}^{k+i} \leq S\beta_{\min}^{-n} \frac{\left(\frac{e^2 n^2}{k^2}\right)^k}{\left(\frac{n}{k+i}\right)^k} o_{\max}^{k+i} \leq S\beta_{\min}^{-n} \left(\frac{e^2 n(k+i)}{k^2}\right)^k o_{\max}^{k+i} \tag{38}$$

$$\leq S\beta_{\min}^{-n}\left(\frac{e^2 n^2}{k^2}\right)^k O_M = S\beta_{\min}^{-n}\left(\frac{en}{k}\right)^{2k} O_M, \quad \text{where } O_M = \max(o_{\max}^k, o_{\max}^n), \tag{39}$$

$$\text{Var}(\hat{M}_k^B) = \binom{n}{k}^2 \text{Var}\left(\sum_{i=1}^{n-k}(-1)^{i-1}\frac{\Phi_{k+1}}{\binom{n}{k+1}}\right) \tag{40}$$

$$\leq (n-k)\binom{n}{k}^2 \text{Var}\left(\frac{\Phi_{k+1}}{\binom{n}{k+1}}\right) \tag{41}$$

$$\leq S(n-k)\left(\frac{en}{k}\right)^{2k} \cdot \beta_{\min}^{-n} O_M = \mathcal{O}(n^{2k+1}) \cdot \beta_{\min}^{-n} O_M, \tag{42}$$

where equation 41 follows from Cauchy-Schwarz inequality $(\text{Var}(\sum_{j=1}^{M} X_i) \leq M \cdot \sum_{j=1}^{M} \text{Var}(X_i))$. The proof follows from dividing the variance of the estimator into two cases: $o_{\max} < 1$ and $o_{\max} > 1$: If $o_{\max} < 1$, $O_M = o_{\max}^k$, and $\text{Var}(\hat{M}_k^B) = \mathcal{O}(n^{2k+1}\beta_{\min}^{-n})$. If $o_{\max} > 1$, $O_M = o_{\max}^n$, and, $\text{Var}(\hat{M}_k^B) = \mathcal{O}(n^{2k+1}\beta_{\min}^{-n} o_{\max}^n)$. ☐

Therefore, the variance exponentially decreases with $n$ if $p_{\max} < 0.5$ or $\frac{1-p_{\max}}{p_{\max}(1-p_{\min})} < 1$.

**Corollary B.5.** *There exists a constant $c > 1$ such that*

$$\text{MSE}(\hat{M}_k^B) \leq \mathcal{O}(n^{2k+1}c^{-n}).$$

*Proof.* From Equ. (12) in the manuscript, the bias $|\mathbb{E}(\hat{M}_k^B) - M_k| \leq S \cdot p_{\max} \cdot n^k \cdot p_{\max}^n$. The proof follows from the fact that $\text{MSE} = \text{Var} + \text{Bias}^2$ and the bound of the variance and the bias. ☐

# C  CONSTRAINTS FOR THE COEFFICIENTS OF THE $\mathbb{E}[M_k]$ REPRESENTATIONS IN THE SEARCH SPACE

In this section, we proof that any coefficients $\alpha_{i,j}$ such that $\sum_{i=1}^{n+1}\sum_{j=i}^{n+1}\alpha_{i,j}g_i(j) = \mathbb{E}[M_k] = \frac{k+1}{n+1}f_{k+1}(n+1) = \binom{n}{k}g_{k+1}(n+1)$ should satisfy the following constraints:

$$\sum_{i=1}^{n+1}\sum_{j=i}^{n+1} c_{i,j}\alpha_{i,j} = \begin{cases} \binom{n}{k} & \text{if } k' = k+1, \\ 0 & \text{otherwise,} \end{cases} \quad \text{where } c_{i,j} = \begin{cases} \binom{n+1-j}{k'-i} & \text{if } 0 \leq k'-i \leq n+1-j, \\ 0 & \text{otherwise,} \end{cases} \tag{43}$$

and vice versa.

*Proof.* Assume the coefficients $\{\alpha_{i,j}\}_{1 \leq i \leq j \leq n+1}$ of the representation $r$ satisfy $\sum_{i=1}^{n+1}\sum_{j=i}^{n+1}\alpha_{i,j}g_i(j) = \mathbb{E}[M_k]$. The proof starts by recursively applying the identity $g_i(j) = g_i(j+1) + g_{i+1}(j+1)$ to the linear combination of the coefficients from $j = 1$ to $n$ transmiting the coefficients to the downward (to the direction of increasing $n$) until all the coefficients of $g_i(j)$ where $j \leq n$ become zero. The resulting coefficients $r' = \{\alpha'_{i,j}\}_{1 \leq i \leq j \leq n+1}$ is the following linear combination of the coefficients of $r$:

$$\alpha'_{i,j} = 0 \quad \text{if } j \leq n, \quad \text{otherwise,} \quad \alpha'_{i,n+1} = \sum_{i'=1}^{n+1}\sum_{j'=i'}^{n+1} c_{i,j}\alpha_{i',j'}, \tag{44}$$

where $c_{i,j}$ is defined as in Equ. (43). This is because $\alpha_{i',j'}$, the coefficient of $g_{i'}(j')$ in $r$ is transmitted to $\alpha'_{i',n+1}$, the coefficient of $g_{i'}(n+1)$ in $r'$ as many times as the number of *propagation paths* from $g_{i'}(j')$ to $g_{i'}(n+1)$ through the identity $g_i(j) = g_i(j+1) + g_{i+1}(j+1)$. Everytime the coefficient is transmitted, the coefficient moves either to the down ($k$ increases by 1) or to the down and right (both $n$ and $k$ increase by 1) in the 2×2 matrix. Therefore, the number of propagation paths from $g_{i'}(j')$ to $g_i(n+1)$ is equal to $\binom{n+1-j'}{i-i'}$, choosing $i-i'$ times to change $k$ from $i'$ to $i$ among $n+1-j'$ steps.

$$r_0 = \left\{ \alpha_{i,j} = \begin{cases} \binom{n}{k} & \text{for } i = k+1 \text{ and } j = n+1 \\ 0 & \text{otherwise.} \end{cases} \right\}$$

The resulting coefficients of $r'$ from $r$ is in fact the same as the coefficients of the initial representation $r_0$ due to the following reason: the final sum of the coefficients of $r'$ becomes $\sum_{k=1}^{n+1} \alpha_{k,n+1} g_k(n+1) = \mathbb{E}[M_k] = \binom{n}{k} g_{k+1}(n+1)$. This is true for any set of probabilities $\langle p_1, \ldots, p_S \rangle$ and $n$. Since, $\binom{n+1}{k+1} p^{k+1} (1-p)^{n-k}$ for $0 \le k \le n$ forms the basis, i.e., the $(n+2)$ Bernstein basis polynomial of degree $n+1$, for the vector space of polynomials of degree at most $n+1$ with real coefficients, the only possible coefficients of the $\mathbb{E}[M_k]$ representations where all the coefficients of $g_i(j)$ where $j \le n$ become zero is the same as the coefficients of the $r_0$ for any set of probabilities $\langle p_1, \ldots, p_S \rangle$ and $n$ due to the linear independence of the basis. Therefore, the constraints in Equ. (43) are necessary for the coefficients of the $\mathbb{E}[M_k]$ representations in the search space.

Next, we show that any representation $r$ that satisfies the constraints in Equ. (43) is a valid representation of the $\mathbb{E}[M_k]$. The proof is straightforward by reversing the above process. The sequence of identities from the above process to reach $r_0$ is reversible to reach $r$ from $r_0$. Therefore, the constraints in Equ. (43) are both necessary and sufficient for the coefficients of the $\mathbb{E}[M_k]$ representations in the search space. $\quad\square$

Following the above proof, any representation of the $\mathbb{E}[M_k]$ that is driven by the four identities,

$$\begin{aligned} \alpha_{i,j} \cdot g_i(j) &= \alpha_{i,j} \cdot ((1-\delta) g_i(j) + \delta g_i(j)) \\ &= \alpha_{i,j} \cdot (g_i(j+1) + g_{i+1}(j+1)) \\ &= \alpha_{i,j} \cdot (g_i(j-1) - g_{i+1}(j)) \\ &= \alpha_{i,j} \cdot (g_{i-1}(j-1) + g_{i-1}(j)), \end{aligned}$$

from the initial representation $r_0$ satisfies the constraints in Equ. (43) and is a valid representation of the $\mathbb{E}[M_k]$, i.e., $\sum_{i=1}^{n+1} \sum_{j=i}^{n+1} \alpha_{i,j} g_i(j) = \mathbb{E}[M_k]$, and vice versa.

## D    COMPUTING THE VARIANCE AND THE MSE OF THE EVOLVED ESTIMATORS

The remaining part to compute the MSE of the evolved estimator $\hat{M}_k^{\text{Evo}}$ is 1) the variance of the missing mass $M_k$, 2) the variance of the evolved estimator $\hat{M}_k^{\text{Evo}}$, and 3) the covariance between the evolved estimator $\hat{M}_k^{\text{Evo}}$ and the missing mass $M_k$.

### D.1    VARIANCE OF $M_k$

$$\text{Var}(M_k) = \text{Var}\left( \sum_x p_x \mathbf{1}(N_x = k) \right) \tag{45}$$

$$= \sum_x p_x^2 \text{Var}(\mathbf{1}(N_x = k)) + \sum_{x \ne y} p_x p_y \text{Cov}(\mathbf{1}(N_x = k), \mathbf{1}(N_y = k)) \tag{46}$$

$$\text{Var}(\mathbf{1}(N_x = k)) = \mathbb{E}\left[ \mathbf{1}(N_x = k)^2 \right] - \mathbb{E}[\mathbf{1}(N_x = k)]^2 = \mathbb{E}[\mathbf{1}(N_x = k)] - \mathbb{E}[\mathbf{1}(N_x = k)]^2 \tag{47}$$

$$= \binom{n}{k} p_x^k (1-p_x)^{n-k} - \left( \binom{n}{k} p_x^k (1-p_x)^{n-k} \right)^2 \tag{48}$$

$$\mathrm{Cov}\left(\mathbf{1}(N_x = k), \mathbf{1}(N_y = k)\right)$$

$$= \mathbb{E}\left[\mathbf{1}(N_x = k)\mathbf{1}(N_y = k)\right] - \mathbb{E}\left[\mathbf{1}(N_x = k)\right]\mathbb{E}\left[\mathbf{1}(N_y = k)\right]$$

$$= \begin{cases} \frac{n!}{k!^2(n-2k)!}p_x^k p_y^k(1-p_x-p_y)^{n-2k} - \binom{n}{k}^2 p_x^k(1-p_x)^{n-k}p_y^k(1-p_y)^{n-k} & \text{if } n \geq 2k, \\ -\binom{n}{k}^2 p_x^k(1-p_x)^{n-k}p_y^k(1-p_y)^{n-k} & \text{otherwise.} \end{cases}$$

## D.2 VARIANCE OF THE EVOLVED ESTIMATOR

Same as $\hat{M}_k^B$, the evolved estimators from the genetic algorithm are also linear combinations of $\Phi_k(n)$s (while varying both $k$ and $n$ unlike $\hat{M}_k^B$). Given the evolved estimator $\hat{M}_k^{\mathrm{Evo}} = \sum_i c_i \Phi_{k_i}(n_i)$, the expected value of $\hat{M}_k^{\mathrm{Evo}}$ is given by substituting $\Phi_k(n)$ with $f_k(n)$:

$$\mathbb{E}(\hat{M}_k^{\mathrm{Evo}}) = \sum_i c_i f_{k_i}(n_i). \tag{49}$$

Given the multinomial distribution $p$, the covariance between $\Phi_k(n)$ and $\Phi_{k'}(n')$, which is needed to compute the variance of $\hat{M}_k^{\mathrm{Evo}}$ as Equ. (25), can be computed as follows:

**Theorem D.1.** *Given the multinomial distribution $p = (p_1, \ldots, p_S)$ with support size $S$, let $X^{n_{total}}$ be the set of $n_{total}$ samples from $p$. Let $X^n$ and $X^{n'}$ be the first $n$ and $n'$ samples from $X^{n_{total}}$, respectively; WLOG, we assume $1 \leq n' \leq n \leq n_{total}$. Then, the covariance of $\Phi_k(n) = \Phi_k(X^n)$ and $\Phi_{k'}(n') = \Phi_{k'}(X^{n'})$ $(1 \leq k \leq n, 1 \leq k' \leq n')$ is given by following:*

$$\mathrm{Cov}\left(\Phi_k(n), \Phi_{k'}(n')\right) = \mathbb{E}\left[\Phi_k(n) \cdot \Phi_{k'}(n')\right] - f_k(n) \cdot f_{k'}(n') \tag{50}$$

$$= \mathbb{E}\left[\left(\sum_x \mathbf{1}(N_x = k)\right) \cdot \left(\sum_{x'} \mathbf{1}(N'_{x'} = k')\right)\right] - f_k(n) \cdot f_{k'}(n') \tag{51}$$

$$= \sum_x \sum_{x'} \mathbb{E}\left[\mathbf{1}(N_x = k \wedge N'_{x'} = k')\right] - f_k(n) \cdot f_{k'}(n'), \tag{52}$$

*where $N'_{x'}$ is the number of occurrences of $x'$ in $X^{n'}$. Depending on the values of $n, n', k, k', x,$ and $x'$, the $\mathbb{E}\left[\mathbf{1}(N_x = k \wedge N'_{x'} = k')\right]$ can be computed as Table 5.*

Table 5: $\mathbb{E}\left[\mathbf{1}(N_x = k \wedge N'_{x'} = k')\right]$ for $\mathrm{Cov}\left(\Phi_k(n), \Phi_{k'}(n')\right)$

| $\forall n, n'$ s.t. | $\forall x, x'$ s.t. | $\forall k, k'$ s.t. | $\mathbb{E}\left[\mathbf{1}(N_x = k \wedge N'_{x'} = k')\right]$ |
|---|---|---|---|
| $n = n'$ | $x = x'$ | $k = k'$ | $\binom{n}{k}p_x^k(1-p_x)^{n-k}$ |
| | | $k \neq k'$ | 0 (*infeasible*) |
| | $x \neq x'$ | $k + k' \leq n$ | $\frac{n!}{k!k'!(n-k-k')!}p_x^k p_{x'}^{k'}(1-p_x-p_{x'})^{n-k-k'}$ |
| | | $k + k' > n$ | 0 (*infeasible*) |
| $n \neq n'$ | $x = x'$ | $k' \leq k$ | $\binom{n'}{k'}p_x^{k'}(1-p_x)^{n'-k'} \cdot \binom{n-n'}{k-k'}p_x^{k-k'}(1-p_x)^{(n-n')-(k-k')}$ |
| | | $k' > k$ | 0 (*infeasible*) |
| | $x \neq x'$ | $k + k' \leq n$ | $\sum_{i=\max(0,k-(n-n'))}^{\min(k,n-k')} \frac{n'!}{k'!i!(n'-k'-i)!} \frac{(n-n')!}{(k-i)!((n-n')-(k-i))!}p_{x'}^{k'}p_x^{k'}(1-p_x-p_{x'})^{n'-k'-i}p_x^k(1-p_x)^{(n-n')-(k-i)}$ |
| | | $k + k' > n$ | 0 (*infeasible*) |

*Proof.* The proof is straightforward from the definition of $N_x$ and $N'_{x'}$. $\square$

## D.3 COVARIANCE BETWEEN THE EVOLVED ESTIMATOR AND THE MISSING MASS

Note that $\mathrm{Cov}\left(\sum_i c_i X_i, Y\right) = \sum_i c_i \mathrm{Cov}\left(X_i, Y\right)$ for any random variables $X_i$ and $Y$. Therefore, the covariance between the evolved estimator $\hat{M}_k^{\mathrm{Evo}} = \sum_i c_i \Phi_{k_i}(n_i)$ and the missing mass $M_k$ is given by $\mathrm{Cov}\left(\hat{M}_k^{\mathrm{Evo}}, M_k\right) = \sum_i c_i \mathrm{Cov}\left(\Phi_{k_i}(n_i), M_k\right)$. Then, again, $\Phi_{k_i}(n_i) = \sum_x \mathbf{1}(N'_x = k_i)$, where $N'_x$ is the number of occurrences of $x$ in $X^{n_i}$, and $M_k = \sum_y p_y \mathbf{1}(N_y = k)$. Therefore, the covariance can be computed given $\mathrm{Cov}\left(\mathbf{1}(N'_x = k_i), \mathbf{1}(N_y = k)\right)$. $\mathrm{Cov}\left(\mathbf{1}(N'_x = k_i), \mathbf{1}(N_y = k)\right) =$

$\mathbb{E}\left[\mathbf{1}(N'_x = k_i \wedge N_y = k)\right] - \mathbb{E}\left[\mathbf{1}(N'_x = k_i)\right]\mathbb{E}\left[\mathbf{1}(N_y = k)\right]$, where the first term is already computed in Table 5, and the second term is $\binom{n_i}{k_i}p_x^{k_i}(1-p_x)^{n_i-k_i} \cdot \binom{n}{k}p_y^k(1-p_y)^{n-k}$.

Having all the necessary components, the MSE of the evolved estimator $\hat{M}_k^{\text{Evo}}$ naturally follows.

## E    DETAILS OF THE GENETIC ALGORITHM

---

**Algorithm 1** Genetic Algorithm

---

**Input:** Target frequency $k$, Sample $X^n$, Iteration limit $G$, mutant size $m$
1: Population $P_0 = \{r_0\}$
2: Fitness $f^{\text{best}} = f_0 = fitness(r_0)$
3: Limit $G_L = G$
4: **for** $g$ from 1 to $G_L$ **do**
5:     $P = selectTopM(P_{g-1}, m)$
6:     $P' = lapply(P, mutate)$
7:     $P_g = P' \cup \{r_0\} \cup selectTopM(P_{g-1}, 3)$
8:     $f_g = \min(lapply(P_g, fitness))$
9:     **if** $(g = G_L) \wedge ((f_g = f_0) \vee (f^{\text{best}} > 0.95 \cdot f_g))$ **then**
10:         $G_L = G_L + G$
11:         $f^{\text{best}} = f_g$
12: Estimator $\hat{M}_k^{\text{Evo}} = instantiate(selectTopM(P_{G_L}, 1))$
**Output:** Minimal-MSE Estimator $\hat{M}_k^{\text{Evo}}$

---

Algorithm 1 shows the general procedure of the genetic algorithm (GA) for discovering the estimator $\hat{M}_k^{\text{Evo}}$ with minimal MSE for the probability mass $M_k$. Given a target frequency $k$ (incl. $k = 0$), the sample $X^n$, an iteration limit $G$, and the number $m$ of candidate representations to be mutated in every iteration, the algorithm produces an estimator $\hat{M}_k^{\text{Evo}}$ with minimal MSE. Starting from the *initial representation* $r_0$ (Eqn. (1); Line 1), our GA iteratively improves a population of candidate representations $P_g$, called *individuals*. For every generation $g$ (Line 4), our GA selects the $m$ fittest individuals from the previous generation $P_{g-1}$ (Line 5), mutates them (Line 6), and creates the current generation $P_g$ by adding the initial representation $r_0$ and the Top-3 individuals from the previous generation (Line 7). The initial and previous Top-3 individuals are added to mitigate the risk of convergence to a local optimum. To *mutate* a representation $r$, our GA (i) chooses a random term $r$, (ii) applies the first identity in Sec. C where $\delta$ is chosen uniformly at random, (iii) applies one of the rest of the identities, and (iv) adjusts the coefficients for the resulting representation $r'$ accordingly. The iteration limit $G_L$ is increased if the current individuals do *not* improve on the initial individual $r_0$ or *substantially* improve on those discovered recently (Line 9–12).

### E.1    HYPERPARAMETERS

For evaluating Algorithm 1, we use the following hyperparameters:

- Same as the Orlitsky's study (Orlitsky & Suresh, 2015), which assess the performance of the Good-Turing estimator, we use the hybrid estimator $\hat{p}$ of the empirical estimate and the Good-Turing estimate to approximate the underlying distribution $\{p_x\}_{x \in \mathcal{X}}$ for estimating the MSE of the evolved estimator. The hybrid estimator $\hat{p}$ is defined as follows: If $N_x = k$,

$$\hat{p}_x = \begin{cases} c \cdot \frac{k}{N} & \text{if } k < \Phi_{k+1}, \\ c \cdot \frac{\hat{M}_k^G}{\Phi_k} & \text{otherwise,} \end{cases} \tag{53}$$

  where $c$ is a normalization constant such that $\sum_{x \in \mathcal{X}} \hat{p}_x = 1$.

- The number of generations $G = 100$. To avoid the algorithm from converging to a local minimum, we limit the maximum number of generations to be 2000.

- The mutant size $m = 40$.

- When selecting the individuals for the mutation, we use *tournament* selection with tournament size $t = 3$, i.e., we randomly choose three individuals with replacement and select the best one, and repeat this process $m$ times.

- When choosing the top three individuals when constructing the next generation, we use *elitist* selection, i.e., choosing the top three individuals with the smallest fitness values.

- To avoid the estimator from being too complex, we limit the maximum number of terms in the estimator to be 20.

The actual script implementing Algorithm 1 can be found at the publically available repository
`https://github.com/niMgnoeSeeL/UnseenGA`.

## F  ADDITIONAL EXPERIMENTAL RESULTS

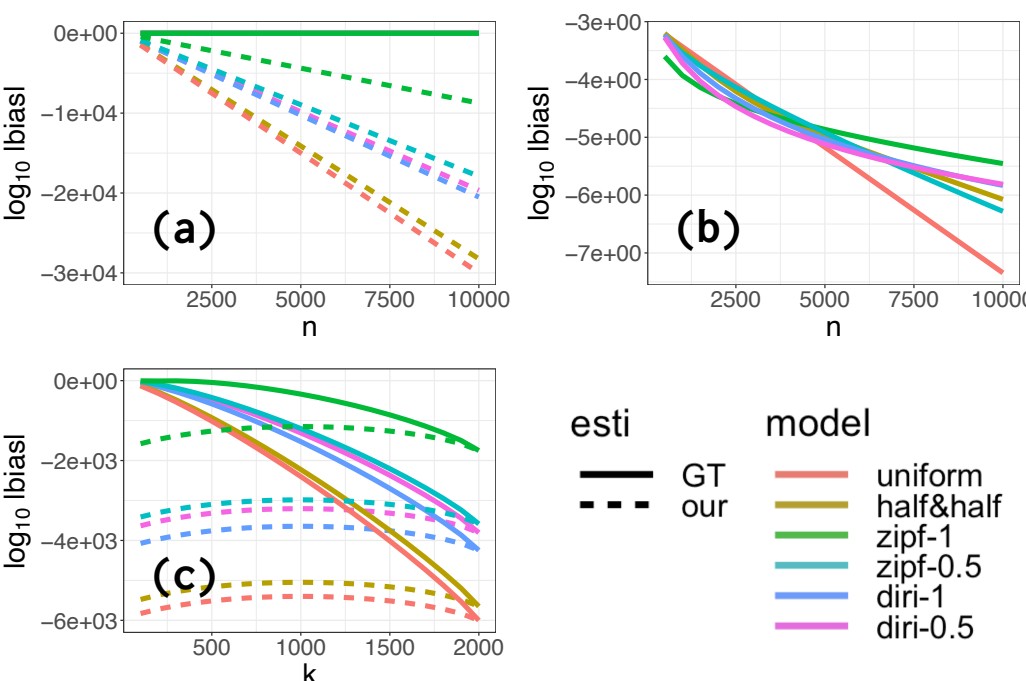

Figure 4: Absolute bias of $\hat{M}_0^B$ and $\hat{M}_0^G$ (a,b) as a function of $n$ for $k = 0$ and (c) as a function of $k$ for $n = 2000$ ($S = 1000$, log-scale).

Figure 4(a) also illustrates the *exponential decay* of our estimator in terms of $n$ and how our estimator is less biased than GT by an exponential factor. In Figure 4(b), we can observe that GT's bias also decays exponentially, although not nearly at the rate of our estimator.

In terms of distributions, a closer look at the performance differences confirms our suspicion that the bias of our estimator is strongly influenced by the probability $p_{\max}$ of the most abundant element, while the bias of GT is strongly influenced by the probability $p_{\min}$ of the rarest element. In fact, by Eqn. (12) the absolute bias of our estimator is minimized when $p_{\max}$ is minimized. By Eqn. (11), GT's bias is minimized if $p_{\min}$ is maximized. Since both is true for the uniform, both estimators exhibit the lowest bias for the uniform across all six distributions. GT performs similar on all distributions apart from the uniform (where bias seems minimal) and zipf-1 (where bias is maximized). For our estimator, if we ranked the distributions by values of $p_{\max}$ with the smallest value first ⟨uniform, half&half, zipf-0.5, zipf-1⟩, we would arrive at the same ordering in terms of performance of our estimator as shown in Figure 4(a).

Table 6: The MSE of the Good-Turing estimator $\hat{M}_0^G$ and the best evolved estimator $\hat{M}_0^{\text{Evo}}$ and for the missing mass $M_0$, the success rate $\hat{A}_{12}$ of the evolved estimator ($X_2$) against the Good-Turing estimator ($X_1$), and the ratio (Ratio, $MSE(M_0^{\text{Evo}})/MSE(\hat{M}_0^G)$) for two support sizes $S = 100$ and $200$, three sample sizes $n$ and six distributions.

| $S$ | $n/S$ | Distribution | $MSE(\hat{M}_0^G)$ | $MSE(\hat{M}_0^{\text{evo}})$ | $\hat{A}_{12}$ | Ratio |
|---|---|---|---|---|---|---|
| 100 | 0.5 | uniform | 1.09e-02 | 7.94e-03 | 0.88 | 72% |
| | | half&half | 1.14e-02 | 7.16e-03 | 0.90 | 63% |
| | | zipf-0.5 | 8.09e-03 | 7.37e-03 | 0.87 | 91% |
| | | zipf-1 | 1.08e-02 | 8.13e-03 | 0.91 | 75% |
| | | diri-1 | 1.10e-02 | 7.97e-03 | 0.92 | 72% |
| | | diri-0.5 | 9.90e-03 | 8.02e-03 | 0.87 | 81% |
| | 1.0 | uniform | 6.05e-03 | 4.29e-03 | 0.97 | 70% |
| | | half&half | 5.46e-03 | 4.07e-03 | 0.98 | 74% |
| | | zipf-0.5 | 3.42e-03 | 3.04e-03 | 0.89 | 88% |
| | | zipf-1 | 5.23e-03 | 4.16e-03 | 0.96 | 79% |
| | | diri-1 | 4.36e-03 | 3.47e-03 | 0.92 | 79% |
| | | diri-0.5 | 3.47e-03 | 2.86e-03 | 0.88 | 82% |
| | 2.0 | uniform | 1.93e-03 | 1.73e-03 | 0.96 | 89% |
| | | half&half | 1.57e-03 | 1.42e-03 | 0.93 | 90% |
| | | zipf-0.5 | 1.26e-03 | 1.08e-03 | 0.94 | 85% |
| | | zipf-1 | 1.73e-03 | 1.54e-03 | 0.97 | 88% |
| | | diri-1 | 1.23e-03 | 1.05e-03 | 0.91 | 85% |
| | | diri-0.5 | 9.41e-04 | 8.08e-04 | 0.86 | 85% |
| 200 | 0.5 | uniform | 5.44e-03 | 4.23e-03 | 0.90 | 77% |
| | | half&half | 5.65e-03 | 4.28e-03 | 0.92 | 75% |
| | | zipf-0.5 | 3.79e-03 | 3.64e-03 | 0.77 | 96% |
| | | zipf-1 | 5.29e-03 | 3.98e-03 | 0.95 | 75% |
| | | diri-1 | 5.45e-03 | 4.01e-03 | 0.92 | 73% |
| | | diri-0.5 | 4.95e-03 | 4.05e-03 | 0.86 | 82% |
| | 1.0 | uniform | 3.01e-03 | 2.46e-03 | 0.98 | 81% |
| | | half&half | 2.73e-03 | 2.26e-03 | 0.96 | 82% |
| | | zipf-0.5 | 1.61e-03 | 1.47e-03 | 0.88 | 91% |
| | | zipf-1 | 2.58e-03 | 2.21e-03 | 0.96 | 85% |
| | | diri-1 | 2.18e-03 | 1.92e-03 | 0.84 | 87% |
| | | diri-0.5 | 1.74e-03 | 1.56e-03 | 0.79 | 89% |
| | 2.0 | uniform | 9.67e-04 | 9.21e-04 | 0.99 | 95% |
| | | half&half | 7.88e-04 | 7.37e-04 | 0.98 | 93% |
| | | zipf-0.5 | 6.01e-04 | 5.66e-04 | 0.91 | 94% |
| | | zipf-1 | 8.66e-04 | 7.99e-04 | 1.00 | 92% |
| | | diri-1 | 6.15e-04 | 5.56e-04 | 0.86 | 90% |
| | | diri-0.5 | 4.71e-04 | 4.25e-04 | 0.80 | 90% |

Table 6 shows the MSE of the Good-Turing estimator $\hat{M}_0^G$ and the best evolved estimator $\hat{M}_0^{\text{Evo}}$ for the missing mass $M_0$, the success rate $\hat{A}_{12}$ of the evolved estimator ($X_2$) against the Good-Turing estimator ($X_1$), and the ratio (Ratio, $MSE(M_0^{\text{Evo}})/MSE(\hat{M}_0^G)$) for two support sizes $S = 100$ and $200$, three sample sizes $n$ and six distributions.

We evaluated the variance of the MSE of the evolved estimator $\hat{M}_0^{\text{Evo}}$ from GA due to its randomness. Using $(S, n) = (100, 100)$, we tested two distributions (uniform and zipf-1) on five sample datasets, running the GA 20 times per dataset. Table 7 shows the mean, median, and variance of the MSE for the evolved estimator as well as the MSE for the Good-Turing estimator for comparison (in parentheses). It shows that the variance of the evolved estimators' MSE is small, indicating the GA's stability and the robustness of the evolved estimator, which consistently outperforms the Good-Turing estimator.

Table 7: Variance of the evolved estimator $\hat{M}_0^{\text{Evo}}$.

| Dataset Index | Uniform (MSE of Good-Turing = 6.0e-3) | | | Zipf (MSE of Good-Turing = 3.4e-3) | | |
|---|---|---|---|---|---|---|
| | Mean MSE | Median MSE | Variance | Mean MSE | Median MSE | Variance |
| 1 | 3.8e-03 | 3.9e-03 | 2.9e-06 | 2.7e-03 | 2.4e-03 | 1.6e-07 |
| 2 | 5.1e-03 | 5.3e-03 | 5.4e-07 | 2.5e-03 | 2.4e-03 | 7.9e-08 |
| 3 | 5.6e-03 | 5.3e-03 | 1.6e-06 | 2.8e-03 | 2.9e-03 | 2.0e-07 |
| 4 | 3.8e-03 | 3.4e-03 | 2.0e-06 | 3.4e-03 | 3.2e-03 | 2.6e-07 |
| 5 | 6.5e-03 | 5.5e-03 | 4.6e-06 | 2.7e-03 | 2.6e-03 | 8.4e-08 |

Table 8: MSE comparison for the probabiltiy mass estimation, i.e., $k > 0$ (Distribution: uniform, $S = 100$, $n = 100$).

| $k$ | $MSE(\hat{M}_k^G)$ | $MSE(\hat{M}_k^{\text{Evo}})$ |
|---|---|---|
| 1 | 2.3e-03 | 1.1e-03 |
| 2 | 1.9e-03 | 5.7e-04 |
| 3 | 1.0e-03 | 2.6e-04 |
| 4 | 3.5e-04 | 1.7e-04 |

We evaluated the probability mass estimation for $0 < k \leq 4$ for uniform distribution with $S = 200, n = 200$. The result shown below indicates that our method consistently outperforms the Good-Turing estimator across $k$ values. We focus on small $k$, as practical interest often lies in unseen and rare categories where empirical probability estimates are biased.

## G    RECENT RELATED WORK ON ESTIMATING MISSING MASS

In this section, we provide a brief overview of the recent related work on estimating the properties of the underlying distribution from a sample (Painsky, 2023; Valiant & Valiant, 2017; Wu & Yang, 2019); those works can either directly or indirectly be used to estimate the missing mass $M_0$. We first describe each method and how it can be used to estimate the missing mass $M_0$. Then, we conduct experiments comparing the performance of the methods against our estimator $\hat{M}_0^B$ in terms of the mean squared error (MSE).

While our objective is to estimate the missing mass $M_0$, Wu & Yang (2019) focus on estimating the support size $S$ of a multinomial distribution $p$ given a sample $X^n$ of size $n$. However, since there can be arbitrarily many unseen classes in the missing mass, they restrict their analysis to distributions whose $\min(p) \geq 1/k$ for a given constant $k$. This implies, the estimand $S$ is assumed to be upper-bounded by $k$. In their experiments, they compare their estimator $\hat{S}^W$ against the estimator $\hat{S}^G = S(n)/(1 - \hat{M}_0^G)$ where $S(n)$ is the number of observed classes and $\hat{M}_0^G$ is the Good-Turing estimator of the missing mass. Integrating both equations, we can construct an estimator of the missing mass $\hat{M}^W$ from their estimate $\hat{S}^W$ of the support size as $\hat{M}^W = 1 - S(n)/\hat{S}^W$ for comparison with our estimator of the missing mass.[4]

Valiant & Valiant (2017) propose to estimate a "plausible" histogram from the frequencies of frequencies (FoF) and, from that, to compute functionals such as entropy or the support size. These metrics are more complex, higher-level metrics than the raw probability mass/missing mass, and the plausible histogram computed from Algorithm 1 in this paper cannot be used directly to estimate the missing mass. Nevertheless, for the purpose of comparison, we can consider the $1 - \sum_i x_i$ used in their algorithm (Algorithm 1 in Valiant & Valiant (2017)) as the missing mass; $x_i$ is the fine mesh of values that discretely approximate the potential support of the histogram. However, since they are not originally intended to represent the probability mass of the unseen samples, we expect the missing mass estimation to be not as accurate as our method.

---

[4]However, we note that $S(n)/S \neq M_0$ for all distributions, except the uniform. Consider a distribution $p = <0.9, 0.1>$ and assume the first sample returns the first class. Then, $S(n)/S = 0.5$ while $M_0 = 0.1$.

Painsky (2023) considers the problem of finding coefficients $\beta_1, \cdots, \beta_n$ in the missing mass estimator $\hat{M}_0^{\beta} = \sum_{i=1}^n \beta_i \Phi_i$ such that the worst-case $l_2^2$ risk over all possible distributions $\mathcal{P}$ is minimized. Using an upper bound on the estimation risk for a given distribution, they define an algorithm that solves a constrained programming problem to find the first two coefficients $\beta_1$ and $\beta_2$ and thus generate an estimator $\hat{M}_0^{\beta_1, \beta_2}$ of the missing mass with minimized risk across all distributions.

Our work makes several contributions over Painsky (2023).

- We generalize the estimation problem beyond the missing mass $M_0$ to the estimation for the total probability mass $M_k$ across all elements that appear $k$ times in the sample for any value of $k : 0 \leq k \leq n$.

- Our (distribution-free) method searches for the (distribution-specific) estimator, which minimizes the risk for that particular (unknown) distribution given only the sample. We propose a genetic algorithm, define a search space of valid representations of $\mathbb{E}[M_k]$ which is searched, and define a method to instantiate a representation into an estimator that is being evaluated for fitness.

- Our set of estimators is parametric not only in $\Phi_i = \Phi_i(n)$ but also the FoF of subsequences $X^j = \langle X_1, ..., X_j \rangle$ of $X^n$, i.e., we consider $\sum_{i=1}^n \sum_{j=1}^n \beta_{i,j} \Phi_i(j)$ (c.f the dependency between FoF in Figure 1).

We conducted 100 repetitions of the experiment with the same setting as in the main paper can calculated the MSE of each method. In particular for Wu & Yang (2019)' method, although $\min(p)$ is not normally known, we set $1/k = \min(p)$ for our experiment. Table 9 shows the MSE of each method for two support sizes $S = 100$ and $200$, three sample sizes $n$ and six distributions. The bold values indicate the best performing estimator. Overall, our search-based method outperforms the other three methods in terms of mean squared error (MSE). This is because in contrast to ours, Wu & Yang (2019) and Valiant & Valiant (2017) are not directly designed to estimate the missing mass, and while all are distribution-free estimation methodologies, only ours generates an estimator with a reduced MSE on that *specific* distribution.

- We found that the MSE of Wu & Yang (2019) is roughly 6x higher than that of our method median-wise. Yet, their method produces significantly inaccurate estimations for some cases; the MSE of their method in those cases could be up to six orders of magnitude higher than that of our method. We use the implementation provided by the authors.

- Same as what we expect, we find Valiant & Valiant (2017)'s MSE is orders of magnitude higher than that of our method (median: 128x higher). We use the code from the paper's appendix and convert it from MATLAB code to Python code.

- Our empirical results confirm that our method outperforms Painsky (2023)'s in terms of MSE; it shows that Painsky's method has a higher MSE than that of our method (median: 10%, mean: 14%); although Painsky's method performs better than the Good-Turing estimator (where our method outperforms Good-Turing by 25%) as they intended, our method still outperforms it.

Table 9: The MSE of the missing mass estimators for two support sizes $S = 100$ and $200$, three sample sizes $n$ and six distributions. The bold values indicate the best performing estimator.

| S | n | dist | Chebyshev | Valiant | Painsky | Ours |
|---|---|---|---|---|---|---|
| 100 | 50 | diri-1 | 3.65e+01 | 2.27e-01 | 9.82e-03 | **7.97e-03** |
| | | diri-0.5 | 1.05e+03 | 2.34e-01 | **7.46e-03** | 8.02e-03 |
| | | half&half | **4.67e-03** | 1.43e-01 | 1.01e-02 | 7.16e-03 |
| | | uniform | 9.40e-03 | 1.01e-01 | 7.97e-03 | **7.94e-03** |
| | | zipf-1 | 4.36e-02 | 1.33e-01 | 7.90e-03 | **7.37e-03** |
| | | zipf-0.5 | **4.73e-03** | 1.43e-01 | 8.51e-03 | 8.13e-03 |
| | 100 | diri-1 | 7.68e-02 | 4.28e-01 | **3.45e-03** | 3.47e-03 |
| | | diri-0.5 | 6.02e+00 | 3.84e-01 | **2.63e-03** | 2.86e-03 |
| | | half&half | 8.41e-03 | 4.05e-01 | 5.49e-03 | **4.07e-03** |
| | | uniform | 5.08e-03 | 3.49e-01 | 7.11e-03 | **4.29e-03** |
| | | zipf-1 | 5.87e-02 | 1.64e-01 | 3.40e-03 | **3.04e-03** |
| | | zipf-0.5 | 8.82e-03 | 3.47e-01 | **3.94e-03** | 4.16e-03 |
| | 200 | diri-1 | 8.70e-02 | 6.05e-01 | 1.34e-03 | **1.05e-03** |
| | | diri-0.5 | 1.29e+01 | 5.22e-01 | 1.01e-03 | **8.08e-04** |
| | | half&half | 6.70e-03 | 6.63e-01 | 1.73e-03 | **1.42e-03** |
| | | uniform | 3.89e-03 | 6.78e-01 | 2.27e-03 | **1.73e-03** |
| | | zipf-1 | 4.98e-02 | 1.91e-01 | 1.59e-03 | **1.08e-03** |
| | | zipf-0.5 | 6.24e-03 | 5.60e-01 | 1.73e-03 | **1.53e-03** |
| 200 | 100 | diri-1 | 5.12e-02 | 2.44e-01 | 4.88e-03 | **4.01e-03** |
| | | diri-0.5 | 3.03e+00 | 3.53e-01 | 5.02e-03 | **4.05e-03** |
| | | half&half | **1.88e-03** | 1.73e-01 | 6.16e-03 | 4.28e-03 |
| | | uniform | 9.78e-03 | 1.29e-01 | 6.54e-03 | **4.23e-03** |
| | | zipf-1 | 1.07e-01 | 1.43e-01 | **2.93e-03** | 3.64e-03 |
| | | zipf-0.5 | **2.45e-03** | 1.75e-01 | 4.42e-03 | 3.98e-03 |
| | 200 | diri-1 | 1.15e-01 | 4.88e-01 | 2.01e-03 | **1.92e-03** |
| | | diri-0.5 | 2.28e+00 | 5.08e-01 | 1.75e-03 | **1.56e-03** |
| | | half&half | 5.10e-03 | 4.23e-01 | 2.42e-03 | **2.26e-03** |
| | | uniform | 3.12e-03 | 3.67e-01 | 2.58e-03 | **2.46e-03** |
| | | zipf-1 | 1.16e-01 | 1.76e-01 | **1.17e-03** | 1.47e-03 |
| | | zipf-0.5 | 5.84e-03 | 3.73e-01 | **2.12e-03** | 2.21e-03 |
| | 400 | diri-1 | 5.44e-01 | 6.56e-01 | **5.00e-04** | 5.56e-04 |
| | | diri-0.5 | 1.94e+00 | 6.69e-01 | 4.68e-04 | **4.25e-04** |
| | | half&half | 4.77e-03 | 7.07e-01 | 7.62e-04 | **7.38e-04** |
| | | uniform | 1.48e-03 | 7.07e-01 | **8.63e-04** | 9.21e-04 |
| | | zipf-1 | 8.45e-02 | 2.08e-01 | **5.06e-04** | 5.66e-04 |
| | | zipf-0.5 | 4.94e-03 | 6.00e-01 | 1.03e-03 | **7.99e-04** |

