# OpenReview forum: "How Much is Unseen Depends Chiefly on Information About the Seen"
_ICLR.cc/2025/Conference — ICLR 2025 Spotlight_

### Official Review · Reviewer_Wyr2 · 2024-10-24

**Soundness:** 4
**Presentation:** 4
**Contribution:** 2
**Rating:** 8
**Confidence:** 3

**Summary:**

This paper studies the problem of estimating the probability of having test points that do not appear in the training data. A class of estimators is provided, and the method for finding an optimal estimator is discussed.

**Strengths:**

The paper is well-written: the setting discussed in this work, as well as the proposed methodology, is clearly stated, and relevant questions are answered.

**Weaknesses:**

I don't feel that the motivation for and usefulness of the problem are sufficiently explained.

**Questions:**

1. As stated in the 'weakness', could the authors provide more explanations for why this is an important problem? When is it helpful to know in advance the probability of a test point being equal to one of the training/calibration data points?

2. Specifically, what do the proposed method (or existing methods) advise practitioners to do with their datasets? For example, if one is given the training and calibration data along with test feature inputs, and the goal is to predict the test outcome, there can be various procedures with different approaches---how does the problem/method in this work affect the overall procedure? If we know that the test point is unlikely to be equal to (or drawn from the same distribution as) one of the observed points, what can we do with that information?

3. I'm not sure if it is appropriate to call the proposed method a 'distribution-free' method. According to the footnote, the context in this work is that the method 'does not impose assumptions on the parameters $p$ or $n$,' but this seems more like a standard setting for a parametric (multinomial) model, and is quite different from the usual meaning of 'distribution-free' (such as in conformal prediction context).

---

> ### Author Response · Authors · 2024-11-19
>
> ## **Motivation and usefulness**:
> *Q. Why is this an important problem? And what the outcome of our method suggests for the practitioner to act.*
>
> Thanks for noting the importance of the motivation and usefulness of the problem.
>
> > Why is this an important problem?
>
> Knowing what is missing from the sample is a crucial problem in statistics and machine learning. It is directly associated with the reliability/accuracy of the model; consider the case where previously unseen events are critical (e.g., medical diagnosis, software vulnerability detection). Thus, many studies have been conducted to estimate the properties of the underlying distribution from the sample [1-7]. **The missing mass [4-7] is the most fundamental property among them**; it is the probability mass of the unseen categories.
>
> In machine learning, the **missing mass measures how representative the training data is of the unknown distribution**. If the missing mass is high, the training is not very representative, and a trained classifier is unlikely to predict the correct class. If we manually label training data, the missing mass also measures saturation. We may decide that the labeling effort has been sufficient and saturation has been reached when the missing mass is below a certain threshold.
>
> While the probability of an observed event $x$ can at least be (over-)estimated using the empirical probability $\hat p_x^{Emp}=N_x/n$, the missing mass cannot be estimated directly since the total empirical probability mass over unobserved events is zero by definition, i.e., $0=\sum_{x\in X_0}\hat p_x^{Emp}$ where $X_0=\lbrace x|x\in X\wedge N_x=0 \rbrace$. Thus, existing methods, including the famous Good-Turing estimator, use the (frequencies of) frequencies $f_k$ to estimate the missing mass. In this paper, we 1) **precisely characterize the bias and the error of the well-known Good-Turing estimator in the multinomial distribution**, and 2) **propose a new distribution-free method** to, given only the sample, find the distribution-specific estimator with a **minimized mean-squared error (MSE) outperforming the existing methods**.
>
> > What does the outcome of our method suggest for the practitioner to act?
>
> ***Tl;dr:*** The proposed method allows practitioners to determine whether additional data collection or labeling is necessary to address unseen components of the underlying distribution. Also, it ensures efficient use of resources for data collection while maintaining model reliability, not wasting resources on unnecessary data collection.
>
> Our method provides practitioners with a reliable way to assess the representativeness of their data and identify gaps affecting model reliability. If unseen events are prevalent, practitioners can decide whether additional data collection or labeling is necessary to address them. Conversely, when the model is already reliable, unnecessary data collection can be avoided, enabling efficient use of resources. This is particularly valuable in domains like software vulnerability detection or medical diagnostics, where proactive risk management is critical, and in resource-constrained environments where data collection is expensive or time-consuming.
>
>
> ## **Meaning of distribution-free**
> *Q. "I'm not sure if it is appropriate to call the proposed method a 'distribution-free' method."*
>
> We follow the definition of distribution-free estimation in the established literature on the missing mass estimation problem: Since Good's formulation of the problem [5], missing mass estimation has been defined in the context of drawing N i.i.d. samples from a categorical population, where $p_1+\dots+p_S=1$ and $n_1+\dots+n_S=N$. Here, $p_i$ represents the probability of sampling species $i$, and $n_i$ is the number of occurrences of species $i$. In this context, following Good's terminology [5], we define a distribution-free estimator as one that makes no assumptions about $p$.
>
> - [1] Chao, A. et al., Unveiling the species-rank abundance distribution by generalizing the Good-Turing sample coverage theory. Ecology, 2015
> - [2] J. Acharya, et al., Optimal probability estimation with applications to prediction and classification. Proceedings of Machine Learning Research, 2013
> - [3] Y. Hao & P. Li. Optimal prediction of the number of unseen species with multiplicity. Advances in Neural Information Processing Systems, 2020
> - [4] I. J. Good & G. H. Toulmin. The number of new species, and the increase in population coverage, when a sample is increased. Biometrika, 1956
> - [5] I. J. Good. The population frequencies of species and the estimation of population parameters. Biometrika, 1953
> - [6] A. Painsky. Convergence guarantees for the good-turing estimator. Journal of Machine Learning Research, 2022
> - [7] A. Orlitsky and A. T. Suresh. Competitive distribution estimation: Why is good-turing good. Advances in Neural Information Processing Systems, 2015

---

> > ### Comment · Reviewer_Wyr2 · 2024-11-26
> >
> > I appreciate the authors' comprehensive response. I am not entirely convinced by the use of the term 'distribution-free,' but this is a minor issue, and I would like to keep the score as before.

---

### Official Review · Reviewer_QhE7 · 2024-11-03

**Soundness:** 3
**Presentation:** 3
**Contribution:** 3
**Rating:** 8
**Confidence:** 2

**Summary:**

This paper addresses the challenge of estimating the expected value of  $M_k$ , which represents the probability that the (n + 1)th observation is an element observed exactly $k$  times in the training data. By precisely characterizing the dependencies among frequency counts, the authors provide a detailed decomposition of $ \mathbb{E} [M_k] $. They introduce a class of estimators that can be constructed using algorithm-based optimization. These proposed estimators outperform the well-known Good-Turing estimator in terms of accuracy.

**Strengths:**

This paper is significant, since estimating the missing mass is a classic and fundamental problem in statistics with broad practical applications. Making advances over the widely used Good-Turing estimator is important, and such progress has the potential to bring substantial empirical improvements. This paper is also novel in its approach: their analysis do not rely on Poisson approximation, which allows for deeper and more flexible analysis. Additionally, the paper presents a thorough evaluation of the proposed algorithms, including theoretical insights and synthetic experiments that demonstrate the minimal-bias estimator’s substantially lower bias compared to the Good-Turing estimator across various distributions.

**Weaknesses:**

No obvious weakness is found.

**Questions:**

None.

---

> ### Author Response · Authors · 2024-11-19
>
> Thank you very much for your strongly positive feedback! We appreciate your supportive comments.

---

### Official Review · Reviewer_JBVY · 2024-11-04

**Soundness:** 3
**Presentation:** 3
**Contribution:** 3
**Rating:** 6
**Confidence:** 4

**Summary:**

The paper makes a contribution to the estimation of the missing mass probability by providing a distribution free estimator that minimizes the mean-squared error after formalizing it as a constrained optimization problem. The authors provide a genetic algorithm that numerically solves the program. They use synthetic and real data experiments on various distributions to compare the resulting estimator to the classically used Good and Turing estimator.

**Strengths:**

- The paper clearly states the problem formulation (fitness function and search space) and proposes a numerical algorithm to solve it.
- The resulting estimator shows a strong performance in terms of MSE and is relatively cheap to compute.
- The presentation and the figures are very clear and well-explained.

**Weaknesses:**

- The empirical application is $\textbf{insufficient}$, only one real world dataset was used and the experiments were done only on 50 datapoints.
- No $\textbf{theoretical guarantees}$ were given for the minimal MSE estimator of the missing mass probability.
- Comparing the proposed genetic Aagorithm to some other optimization approaches  as well as studying the properties of the resulting estimators would be helpful.

**Questions:**

- Are there any theoretical guarantees in terms of MSE when running the optimization on a smaller sample size to compute the estimator for a larger size using the provided formula ?

- How does the proposed minimum MSE estimator perform compared to a distribution-aware estimator for different distributions?

- How does the effectiveness of the proposed estimator look like for strictly positive values of $k$?

- Given a fixed dataset, what is the variance of the resulting estimator's MSE output by the GA (given randomness in mutations)?

---

> ### Author Response · Authors · 2024-11-19
>
> We appreciate your insightful comments and suggestions. We organized our responses to two categories: **Experimental Evaluation** and **Other Comments**.
> ## Experimental Evaluation
> >Q. "What is the variance of the estimator's MSE (given randomness in mutations)?"
>
> ***Done!*** We evaluated the variance of the MSE of the evolved estimator from GA due to its randomness. Using (S,n)=(100,100), we tested two distributions (uniform and Zipf) on five sample datasets, running the GA 20 times per dataset. The table below shows the mean, median and variance of the MSE for the evolved estimator as well as the MSE for the Good-Turing estimator for comparison (in parentheses). It shows that the variance of the evolved estimators' MSE are small, indicating the GA’s stability and the robustness of the evolved estimator, which consistently outperforms the Good-Turing estimator.
>
> - Uniform (MSE of Good-Turing = 6.0e-3)
>
> |Dataset Index|Mean MSE|Median MSE|Variance|
> |-|-|-|-|
> |1|3.8e-03|3.9e-03|2.9e-06|
> |2|5.1e-03|5.3e-03|5.4e-07|
> |3|5.6e-03|5.3e-03|1.6e-06|
> |4|3.8e-03|3.4e-03|2.0e-06|
> |5|6.5e-03|5.5e-03|4.6e-06|
>
> - Zipf (MSE of Good-Turing = 3.4e-3)
>
> |Dataset Index|Mean MSE|Median MSE|Variance|
> |-|-|-|-|
> |1|2.7e-03|2.4e-03|1.6e-07|
> |2|2.5e-03|2.4e-03|7.9e-08|
> |3|2.8e-03|2.9e-03|2.0e-07|
> |4|3.4e-03|3.2e-03|2.6e-07|
> |5|2.7e-03|2.6e-03|8.4e-08|
>
> >Q. "How is the effectiveness of positive values of k?"
>
> ***Done!*** We evaluated the probability mass estimation for $0<k\le 4$ for two distributions (uniform and Zipf) with $S=200,n=200$. The result shown below indicates that our method consistently outperforms the Good-Turing estimator across k values. We focus on small k, as practical interest often lies in unseen and rare categories where empirical probability estimates are biased.
> |k|Good-Turing|Evolved|
> |-|-|-|
> |1|2.3e-3|1.1e-3|
> |2|1.9e-3|5.7e-4|
> |3|1.0e-3|2.6e-4|
> |4|3.5e-4|1.7e-4|
> >"only one real world dataset was used ... on 50 datapoints."
>
> ***Done!*** We applied our method to the Shakespeare dataset [1] ($S=935$, |Dataset|=111,396), commonly used in similar studies [2], focusing on missing mass estimation for player frequency. The MSE comparison below shows that our method consistently outperforms the Good-Turing estimator across all sample sizes.
> |n|Good-Turing|Evolved|
> |-|-|-|
> |100|4.2e-3|3.0e-3|
> |200|2.6e-3|2.1e-3|
> |500|9.0e-4|8.8e-4|
>
> ## Other Comments
>
> >Q. "Are there any theoretical guarantees in terms of MSE when running the optimization on a smaller sample size to compute the estimator for a larger size using the provided formula?"
>
> Eq. 16 (p. 6) in the manuscript represents the MSE of the class of estimator, i.e., the linear combination of the frequencies of frequencies, for the probability mass. The adjusted estimator for a larger sample size also belongs to this class. Therefore, one can get the change of MSE in terms of the change of the sample size directly from Eq. 16.
>
> >Q. "How does the proposed minimum MSE estimator perform compared to a distribution-aware estimator for different distributions?"
>
> There is no known distribution-specific estimator for the probability mass for a specific multinomial distribution. To the best our knowledge, all other works investigates the distribution-free estimator. We believe our work is the first work that generates a distribution-specific estimator for an arbitrary multinomial distribution.
>
> [1] Shakespeare plays, 2016, kaggle.com/datasets/kingburrito666/shakespeare-plays
>
> [2] Efron, B., and Thisted, R., “Estimating the Number of Unseen Species: How Many Words Did Shakespeare Know?”, Biometrika, 1976

---

### Author Response · Authors · 2024-11-19

We thank the reviewers for their positive comments and constructive feedback. We address all concerns and suggestions in this rebuttal. We respond to each reviewer's comments individually, addressing their questions and suggestions.

---

### Meta-Review · Area_Chair_5p41 · 2024-12-19

**Metareview:**

The reviewers reach the consensus that the paper is well-written, tackling challenging missing class probability estimation and provide MSE studies based on a class of estimators. The reviewers has reached a consensus that the contribution is substantial and generally agreed that the paper is well-written. In addition, the authors provides additional results during the discussion and is recommended to include in the manuscript. It is recommended that the paper appears in the ICLR conference.

**Additional Comments On Reviewer Discussion:**

The discussions provides additional clarifications for the reviews.

---

### Decision · Program_Chairs · 2025-01-22

Accept (Spotlight)